# LoDEI: a robust and sensitive tool to detect transcriptome-wide differential A-to-I editing in RNA-seq data

Phillipp Torkler [1], Marina Sauer[2], Uwe Schwartz [3], Selim Corbacioglu[2], Gunhild Sommer [2] & Tilman Heise [2] ✉

RNA editing is a highly conserved process. Adenosine deaminase acting on RNA (ADAR) mediated deamination of adenosine (A-to-I editing) is associated with human disease and immune checkpoint control. Functional implications of A-to-I editing are currently of broad interest to academic and industrial research as underscored by the fast-growing number of clinical studies applying base editors as therapeutic tools. Analyzing the dynamics of A-to-I editing, in a biological or therapeutic context, requires the sensitive detection of differential A-to-I editing, a currently unmet need. We introduce the local differential editing index (LoDEI) to detect differential A-to-I editing in RNA-seq datasets using a sliding-window approach coupled with an empirical q value calculation that detects more A-to-I editing sites at the same false-discovery rate compared to existing methods. LoDEI is validated on known and novel datasets revealing that the oncogene MYCN increases and that a specific small non-coding RNA reduces A-to-I editing.

About 170 modifications of ribonucleotides are known to affect biological functions significantly, and misregulation is associated with a growing number of diseases[1,2]. RNA editing is an RNA modification caused by proteins and can lead to changes in the RNA sequence[3,4]. Engineering and manipulating RNA via guided RNA editing has immense therapeutic value currently gaining momentum, and demands precise identification of altered RNA editing signals[5-8]. The most frequent RNA editing in animals is facilitated by genes of the adenosine deaminase acting on the RNA (ADAR) family. ADAR1 and ADAR2 are specific proteins known to convert adenosine to inosine (A-to-I editing), interpreted as guanosine by the cellular machinery such as the ribosome or spliceosome, and appear as guanosine in next-generation sequencing (NGS)[9-11].

Biological functions of ADAR-induced RNA-editing include the editing of a specific position in a coding sequence, leading to an amino acid substitution potentially affecting protein functionality, and A-to-I editing can lead to alternative splicing, altered miRNA function,

changes in the stability of RNA folds, and nuclear retention of the mRNA. Furthermore, most A-to-I editing occurs within double-stranded RNA (dsRNA) caused by Alu and inverted Alu repeats located primarily in 3' untranslated regions (3'UTRs) and introns and this widespread editing of adenosines in untranslated regions can counteract an immune response caused by endogenous dsRNA[9,12].

Besides the biological importance of A-to-I editing, the clinical significance has been recognized more recently. For example, inflamed endothelial cells show a drastic change in ADAR2-mediated A-to-I editing, in cancer altered regulation of A-to-I editing has been observed, and down-regulation of ADAR triggers activation of the dsRNA sensor, leading to translational activation of the interferon systems, beneficial for immune therapies[12-15]. Moreover, the application of designed base editors for site-specific RNA editing, the therapeutic targeting of ADAR, and the identification of other A-to-I regulatory factors require a robust and sensitive tool to compare the editing between two conditions[16-18].

[1]Faculty of Computer Science, Deggendorf Institute of Technology, Dieter-Görlitz-Platz 1, Deggendorf 94469 Bavaria, Germany. [2]Department for Pediatric Hematology, Oncology and Stem Cell Transplantation, University Hospital Regensburg, Franz-Josef-Strauß-Allee 11, Regensbug 93053 Bavaria, Germany. [3]NGS Analysis Center, University of Regensburg, Universitätsstraße 31, Regensburg 93053 Bavaria, Germany. ✉e-mail: tilman.heise@klinik.uni-regensburg.de

So far, bioinformatic software tools detect putative A-to-I editing by aligning RNA-seq data to a corresponding reference genome, followed by detecting $A \to G$ mismatches likely caused by A-to-I editing. Herein, most tools focus on the detection of single edited A-to-I sites. REDItools takes RNA-seq data as input and reports a table of single genomic positions of G/A ratios that can be filtered for single nucleotide polymorphisms (SNPs) if a corresponding DNA-seq file is provided[19,20]. GIREMI focuses on detecting individual editing sites via a mutual information approach without the need for a sequenced genomic reference[21]. Other publications avoid de novo detection and focus on single editing sites listed in A-to-I editing databases like RADAR and REDIportal[22–24]. Since A-to-I editing is not limited to always appearing at identical sites between different samples, reliable detection of single nucleotides is challenging for any single-site approach[25].

To address the drawbacks of the detection of single sites probably caused by the widespread binding of ADAR1 to dsRNA, alternative approaches like RNAEditor, FLARE, and the Alu editing index (AEI) have been proposed that share the common idea to detect A-to-I editing by analyzing a larger genomic region rather than analyzing single nucleotides[25–27].

All tools focus on detecting putative RNA-editing but do not offer methods for detecting *differential* RNA-editing. Differentially edited A-to-I events are of great interest as these sites are a proxy for biologically relevant regions, especially in scenarios where different experimental conditions are compared. To address the need, JACUSA2 and REDIT introduced specific statistical models for differential A-to-I editing detection at single nucleotides to replace general statistical tests like the *t*-test or Mann–Whitney *U*-test[28–32]. By design, all single-site-specific approaches suffer from widespread editing of ADAR1 that may yield little information at an individual position, requiring many samples to detect transcriptome-wide differential A-to-I editing at single sites reliably and make certain modeling assumptions of the underlying data. In contrast, global approaches like the AEI address the widespread editing problem at the cost of losing all positional information and lacking a proper statistical framework. Hence, tools for the robust detection of differential A-to-I still remain to be a major open task[33].

In contrast to model-based approaches used in JACUSA2 and REDIT, non-parametric approaches are an alternative that does not make any assumptions about the observed data. Large-scale NGS experiments contain millions of mismatch events, enabling non-parametric approaches to estimate null distributions empirically and thereby avoiding introducing any model constraints and allowing the calculation of *q* values directly[34–36].

Here, we present the local differential editing index (LoDEI) to detect differential A-to-I editing in two sets of RNA-seq samples on a transcriptome-wide scale using a non-parametric approach for *q*-value calculation. Compared to other methods, LoDEI detects more differential A-to-I editing at the same false discovery rate (FDR) and finds editing events that have remained undetected. Our window-based approach addresses problems caused by widespread editing but keeps the high resolution of single-site approaches. Applying an empirical approach, LoDEI can detect A-to-I editing even in the comparison of single samples, an unfeasible scenario for site-specific, model-based approaches.

Basic research on human disease and drug discovery projects will benefit from LoDEI's increased sensitivity and accuracy since global values like the AEI are unable to recognize treatment effects when drug candidates influence A-to-I editing in bidirectional ways, and single-site approaches often miss target sites and off-target effects or require a high number of samples to detect editing signals.

## Results
The goal of LoDEI is to detect biologically relevant differences in A-to-I editing between two sets of RNA-seq samples (Fig. 1). To identify

differential A-to-I editing, the detection method of LoDEI can be separated into two high-level steps:

1. A sliding window estimates the A-to-I editing signal for each set and calculates the difference between the editing signals of the two sets.
2. To separate calculated A-to-I editing signal differences caused by true A-to-I events from noise, the same sliding window approach is applied to non-A-to-I mismatches (all non-$A \to G$ mismatches), generating the data to allow calculating *q* values empirically.

This paper proceeds with a precise description of the A-to-I editing signal calculation and *q*-value estimation used by LoDEI, followed by an analysis of A-to-I and non-A-to-I signals in various datasets to demonstrate the general applicability of an empirically *q*-value calculation for differential A-to-I signal detection. Finally, we first assess LoDEI's A-to-I editing detection performance by comparing findings with results of the global A-to-I detection as provided by the AEI, and second, we compare the differential A-to-I editing detection of LoDEI with results from the site-specific differential A-to-I detection tools REDIT and JACUSA2.

### Calculating the A-to-I editing difference between two sets of samples
The general idea to account for the editing behavior of ADAR1 and keeping as much positional information as possible is addressed by estimating A-to-I editing using a sliding window approach. Having two sets of samples of RNA-seq data, LoDEI first estimates the A-to-I editing signal for each sample within a window individually. The mean of the estimated editing signals of a window of all samples of a set is calculated, and the difference between the means of the windows of two sets defines the change in RNA-editing (Fig. 1).

More formally, let $S$ and $S'$ be two sets of samples $s$ of RNA-seq data of two different conditions. The observed mismatch counts $m_{i,s}^{x \to y}$ of nucleotide $x$ to $y$, and the sum of all matches and mismatches $c_{i,s}$ at genomic position $i$ in sample $s$ define the editing ratio

$$r_{i,s}^{A \to G} = \frac{m_{i,s}^{A \to G}}{c_{i,s}}. \tag{1}$$

We then define the editing signal $e_{s,w}$ (Fig. 1b [1]) for a sample $s$ and a sliding window $w$ with genomic start and end positions $k$ and $l$ as

$$e_{s,w}^{A \to G} = \sum_{i=k}^{l} r_{i,s}^{A \to G}. \tag{2}$$

Note, LoDEI uses non-overlapping windows with a default size of 51 nucleotides. Using the editing signals of all samples of a set $S$ for a given window $w$, we can calculate

$$z_{S,w}^{A \to G} = \frac{1}{|S|} \sum_{s \in S} e_{s,w}^{A \to G}, \tag{3}$$

where $z_{S,w}^{A \to G}$ is the mean of the editing signals of all samples of a set and serves as an estimate for the A-to-I editing of set $S$ (Fig. 1b, [2]).

After performing the same calculation for $S'$, the change of the editing signal $\delta_{S,S',w}$ (Fig. 1b, [3]) between sets $S$ and $S'$ is given by

$$\delta_{S,S',w}^{A \to G} = z_{S',w}^{A \to G} - z_{S,w}^{A \to G}, \tag{4}$$

and describes the difference of the A-to-I editing between the sets $S$ and $S'$ for a given window $w$.

Next, we propose an empirical *q*-value estimation for $\delta_{S,S',w}^{A \to G}$ values based on non-$A \to G$ differences to detect true editing events and differentiate those from false positive events. A *q* value gives each $\delta$ value

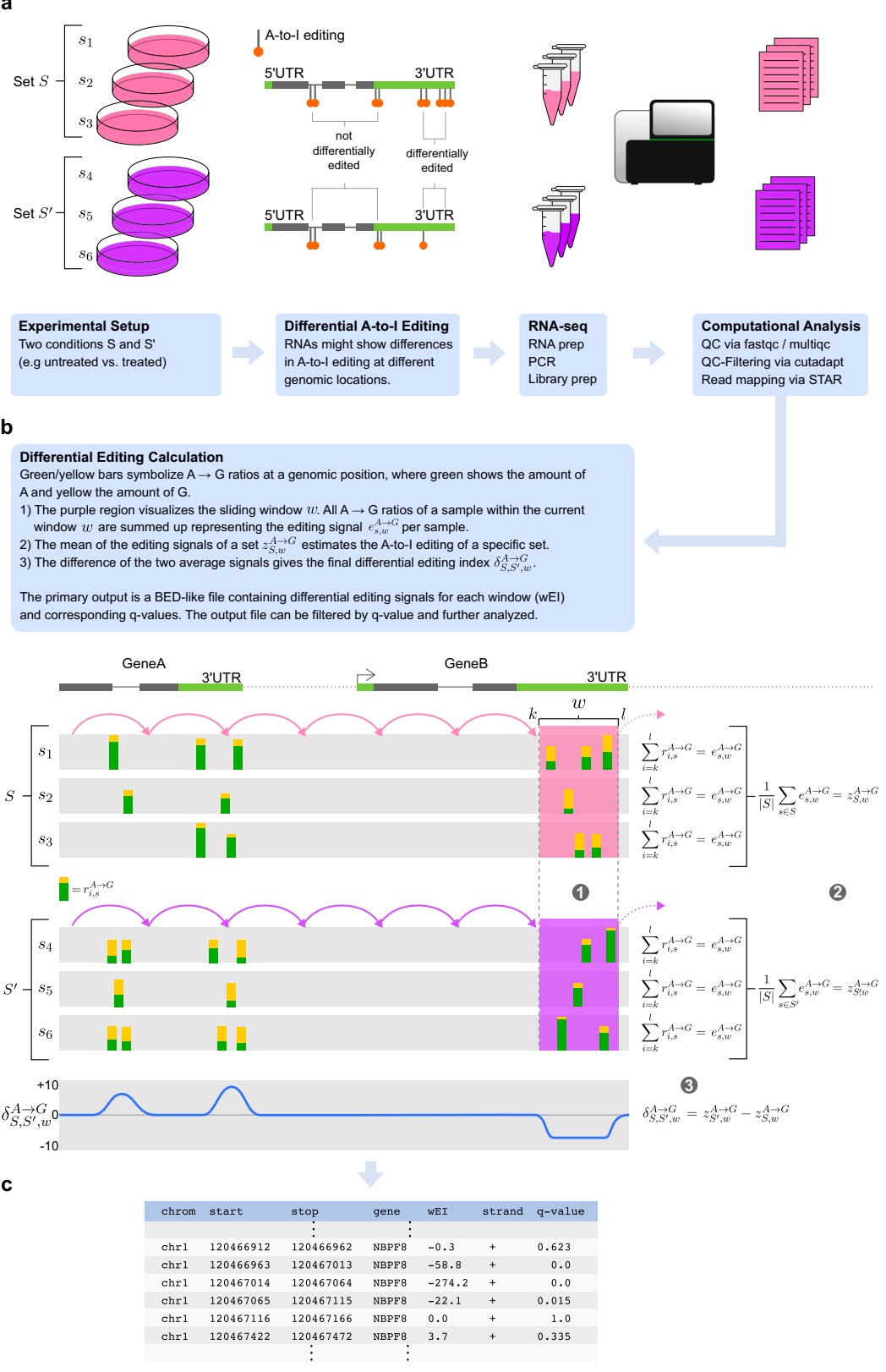

**Fig. 1 | Schematic overview of differential A-to-I editing index calculation. a** Two sets of samples $S$ and $S'$ with different conditions undergo RNA-sequencing followed by standard RNA-seq data analysis of QC checking and read alignment. **b** and **c** Next, differential A-to-I editing signals are calculated by LoDEI's sliding window approach (**b**) yielding the final output table (**c**).

its own individual measure of significance, similar to a $p$ value. A $p$ value is a measure in terms of the false positive rate, whereas the $q$ value is a measure in terms of the false discovery rate and the typical quantity of interest in genome-wide testing[34]. The $q$ value provides a measure of significance for each window in such a way that filtering of windows with a $q$ value ≤ 10% yields an overall FDR of 10% among the filtered windows.

For the $q$ value estimation, we exploit that most non-$A \rightarrow G$ differences are not generated by any editing processes and are likely to derive from various sources like PCR or NGS-induced errors and SNPs. The estimated $q$ value of an A-to-I editing difference $\delta^{A \rightarrow G}$ is the expected proportion of false positives among all differences as or more extreme than the current one. The key to estimating a $q$ value for an observed change of the editing signal is to approximate the number of expected false positives.

Considering $A \rightarrow G$ mismatches as a mixture of A-to-I editing events and noise, the eleven remaining non-$A \rightarrow G$ mismatches (e.g. $A \rightarrow C$, $G \rightarrow A$, etc.) serve as the basis for approximating expected false positives.

Let $D^{x \rightarrow y}$ be the set of $\delta^{x \rightarrow y}$ values calculated by LoDEI, where $x$ and $y$ symbolize a non-$A \rightarrow G$ mismatch used to calculate the differences. First, LoDEI generates an individual set $D^{x \rightarrow y}$ for each non-$A \rightarrow G$ mismatch. Each set approximates the number of expected false positives for a given value of $\delta$ and thus yields an individual $q$ value estimate

$$\hat{q}(\delta, D^{A \rightarrow G}, D^{x \rightarrow y}) = \frac{\#\text{signals} \leq \delta \text{ in } D^{x \rightarrow y}}{\#\text{signals} \leq \delta \text{ in } D^{A \rightarrow G}}, \quad (5)$$

for each non-$A \rightarrow G$ set. The median of the eleven estimated $q$ values is the final estimated $q$ value for a given A-to-I difference $\delta^{A \rightarrow G}$. Note, for $\delta < 0$ the number of signals ≤ $\delta$, and for $\delta > 0$ the number of signals ≥ $\delta$ is used to estimate the number of false positives.

The stronger the calculated $\delta$ values by LoDEI, the smaller the number of those observed differences. This negative correlation can lead to strong variability of the estimated $q$ values if the number of observed $\delta$ values is small. Typically, the $q$ value decreases the stronger the values for $\delta^{A \rightarrow G}$ get. For extreme values of $\delta^{A \rightarrow G}$, where the variability of the $q$ value estimate increases, situations can arise where higher signals get higher $q$ values again, while lower values of $\delta^{A \rightarrow G}$ already had lower $q$ values (Supplementary Fig. 8). These situations are caused by a limited number of available windows and not by underlying biology. Consequently, we assume increasing $q$ values for strong values of $\delta$ is counterintuitive and avoid varying $q$ value estimates by not allowing those estimates to increase again.

Finally, LoDEI reports a BED-like output file including the differential editing signals of all windows, their genomic coordinates, and corresponding $q$ values (Fig. 1c).

## LoDEI confirms known and reveals novel regulators of A-to-I editing

To demonstrate the general applicability of LoDEI's differential editing index calculation and the empirical $q$ value estimation, we analyzed differential A-to-I editing in four different human RNA-seq datasets with 27 samples produced by four different laboratories to cover a broad spectrum of protocols. The first two of the four datasets are known to show differential A-to-I editing and are used to show the general applicability of LoDEI's approach. All datasets show a significant change in the gene expression in at least one of the genes of the ADAR family (Supplementary Table 2).

The first analyzed RNA-seq dataset consisting of two samples per set is known to show a strong reduction of A-to-I editing upon siRNA-induced ADAR1 knockdown (KD) in the glioblastoma cell line U87MG when compared to a control group[16]. Other RNA-binding proteins are known to regulate A-to-I editing besides ADAR, and several publications could show an increase in A-to-I editing upon the

reduction of RO60 (TROVE2)[24,37,38]. Hence, we used the RNA-seq dataset consisting of two control and three knockout samples derived from the RO60 knockout (KO) Lymphoblastoid cell line GM12878 as the second dataset[17].

After evaluating the general applicability of LoDEI on the ADAR KD and RO60 KO datasets, we applied LoDEI to novel datasets to search for differential A-to-I editing.

Across all datasets, a consistent contrast between $A \rightarrow G$ and non-$A \rightarrow G$ differences can be observed. Non-$A \rightarrow G$ differences, such as $G \rightarrow A$, show a different pattern compared to $A \rightarrow G$ differences (Fig. 2 left-column (a, d, g, j) versus middle column (b, e, h, k)). Strong editing differences are almost exclusively observed for $A \rightarrow G$ differences (Fig. 2 middle column (b, e, h, k), orange), and the shape of weak $A \rightarrow G$ differences resembles the shape of non-$A \rightarrow G$ differences represented by $G \rightarrow A$ values, supporting the general applicability of LoDEI's window-based approach to describe differential A-to-I editing and the usage of non-$A \rightarrow G$ differences for the estimation of false positive signals.

To further support LoDEI's general applicability, we analyzed differential A-to-I editing in non-human datasets. We analyzed RNA-seq data from ADAR mutant and wildtype *C. elegans* worms and observed the same consistent contrast between $A \rightarrow G$ and non-$A \rightarrow G$ differences as in the human datasets (Supplementary Fig. 12). Strong editing differences are almost exclusively observed for $A \rightarrow G$ differences.

## LoDEI confirms known regulators of A-to-I editing

In the ADAR1 KD dataset, $A \rightarrow G$ differences calculated by LoDEI show a unidirectional, reduced A-to-I editing in ADAR1 KD cells. Concordantly, $q$ values ≤ 0.1 only appear for negative $\delta^{A \rightarrow G}$ values (Fig. 2c, orange line). Besides $A \rightarrow G$ differences, LoDEI detects a small number of $T \rightarrow C$ mismatches (Supplementary Table 1).

Taken together, LoDEI can calculate differential A-to-I editing and confirms data showing a reduction of A-to-I editing upon ADAR1 knockdown.

Applying LoDEI on the RO60 KO dataset reveals that all $A \rightarrow G$ differences with a $q$ value ≤ 0.1 are exclusively detected with a positive sign (Fig. 2e, f; orange line/points), indicating that A-to-I editing is increased in RO60 KO cells. No other significant changes caused by other types of mismatches are found in this dataset. LoDEI confirms that RO60 represses A-to-I editing and that LoDEI is applicable for identifying factors or conditions regulating A-to-I editing.

To demonstrate potential novel discoveries facilitated by LoDEI's results, we selected a LoDEI window with differential A-to-I editing in the open reading frame of the *SRP9* gene in the RO60 dataset. The analysis of the window in the IGV genome browser revealed two potential silent mutations and one differentially edited site potentially leading in about 30% of the mRNAs to a serine to glycine mutation resulting in an expression of an SRP9 protein isoform (Supplementary Fig. 9).

## LoDEI reveals novel regulators of A-to-I editing

LoDEI reported expected negative as well as positive differential A-to-I editing from published datasets with known negative (ADAR1 KD) and positive effects on A-to-I editing (RO60 KO).

Next, we applied LoDEI on a third dataset using previously published RNA-seq data of different neuroblastoma cell lines, a more challenging scenario compared to the analysis of data from the same cell line[39]. The eight samples were grouped into MYCN-amplified (SK-N-BE(1), LAN-1, IMR-5, CHP-212) and MYCN-non-amplified cell lines (SK-N-AS, SK-N-SH, SH-SY-5Y, SK-N-F1) to analyze the impact of MYCN-amplification (MYCN-amp) on A-to-I editing. First, similar to the ADAR1 KD and RO60 KO datasets, non-$A \rightarrow G$ differences, such as $G \rightarrow A$, show a different pattern compared to $A \rightarrow G$ differences (Fig. 2g, h). All $A \rightarrow G$ differences with a $q$ value ≤ 0.1 are exclusively detected with a positive sign (Fig. 2h, i; orange line/points), highlighting an increased A-to-I

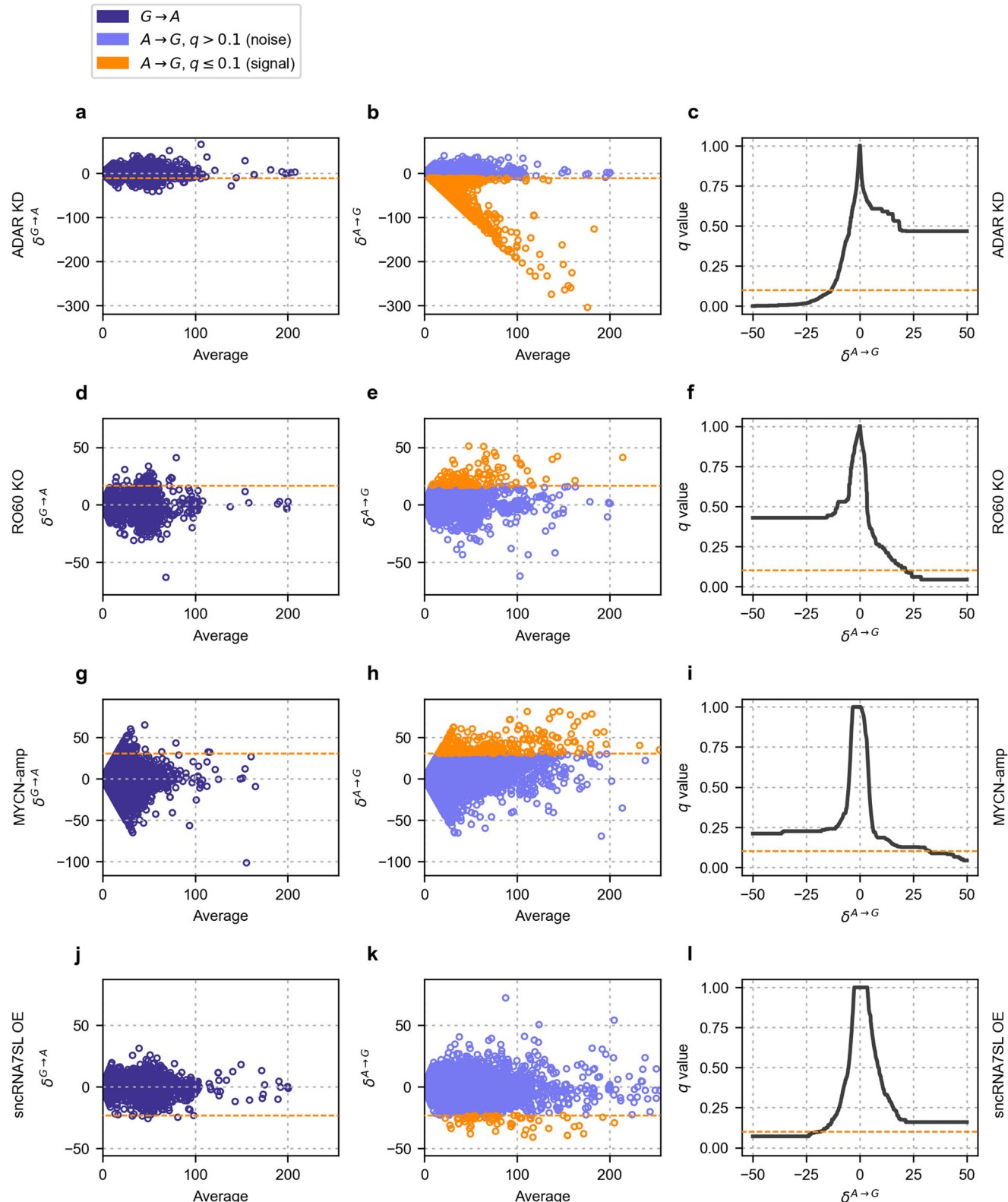

**Fig. 2 | Observed signal differences and empirically derived *q* values.** Rows correspond to the ADAR1 KD (**a**–**c**), RO60 KO (**d**–**f**), MYCN-amp (**g**–**i**), and sncRNA7SL OE (**j**–**l**) datasets. The left and middle columns show Bland–Altman plots[51] of $\delta^{G \to A}$ values considered as noise (left column), and $\delta^{A \to G}$ values considered as a mixture of signal and noise (middle column). The third column shows empirical *q* values. Orange lines show the signal cutoffs derived by LoDEI corresponding to a *q* value ≤ 0.1 in columns 1 and 2 and a *q* value of 0.1 in column 3.

**Table 1 | Comparison of global trends of LoDEI and the Alu editing index**

| Dataset | AEI condition | AEI control | AEI difference | Adjusted p-value | LoDEI #+ | LoDEI #− |
|---|---|---|---|---|---|---|
| ADAR1 KD | 0.31 (2) | 1.48 (2) | −1.17 [−1.20, −1.14] | 0.0006 | 0 | 1624 |
| sncRNA7SL OE | 1.00 (5) | 1.06 (5) | −0.06 [−0.07, −0.05] | 0.00006 | 0 | 64 |
| RO60 KO | 1.16 (3) | 1.07 (2) | 0.09 [−0.06, 0.22] | 0.46 | 114 | 0 |
| MYCN-amp | 1.13 (4) | 0.95 (4) | 0.18 [0.03, 0.36] | 0.14 | 217 | 0 |

Columns 1 and 2 show the mean of the AEI values of all samples of a set for all four datasets and the number of RNA-seq samples (*n*) per set. Column 3 shows the difference between the mean AEI values of columns 1 and 2 and the corresponding 95% confidence interval, and column 4 shows the Benjamini–Hochberg corrected two-sided *t*-test derived *p* values for the comparisons of the AEI means. Columns 5 and 6 show the number of detected windows by LoDEI with *q* values ≤0.1 with increased (#+) and decreased (#−) A-to-I signal. The column 'condition' refers to the ADAR1 KD, sncRNA7SL OE, RO60 KO, and MYCN-amp samples.

editing in MYCN-amp cells. To confirm these findings, we manually inspected detected differential A-to-I edited windows in the IGV genome browser verifying increased A-to-I editing in MYCN-amp cells (Supplementary Fig. 1). Another analysis of a LoDEI window in the MYCN-amp dataset, located in the open reading frame of the *CHAF1A* (Chromatin Assembly Factor 1 Subunit A) gene shows a differential editing site that is also a 3' splice acceptor site. The editing of this site could affect the splicing of the *CHAF1A* mRNA and further suggest that the increase in A-to-I editing associated with MYCN-amplification could have an impact on *CHAF1A* expression (Supplementary Fig. 10). By applying LoDEI, we show for the first time that A-to-I editing is positively regulated in MYCN-amp cell lines.

In the fourth dataset, comprised of 5 replicates per set, we tested the hypothesis that a small non-coding RNA with complementarity to reverse Alu elements modulates A-to-I editing[40]. The neuroblastoma cell line Kelly was transfected with a control RNA or the small non-coding RNA sncRNA7SL (pir-hsa-1254, DQ571003, piR31115) yielding an overexpression of sncRNA7SL (sncRNA7SL OE). $A \rightarrow G$ differences with a *q* value ≤ 0.1 are exclusively detected in the negative direction, demonstrating a unidirectional and reduced A-to-I editing in sncRNA7SL OE cells compared to control-treated cells (Fig. 2k, l; orange line/points, Supplementary Fig. 2). Besides unidirectional $A \rightarrow G$ differences, LoDEI detects more $G \rightarrow C$ differences compared to the number of $A \rightarrow G$ differences in the sncRNA7SL OE dataset (Supplementary Table 1). Taken together, reduced A-to-I editing is detected by LoDEI upon sncRNA7SL transfection.

## Global changes in A-to-I editing identified by LoDEI are supported by the Alu Editing Index

From a global perspective, A-to-I editing differences with *q* values ≤ 0.1 as identified by LoDEI show a unidirectional change in all four datasets (Fig. 2). All windows with a *q* value ≤ 0.1 in the ADAR1 KD and sncRNA7SL OE cells show a decrease, and the RO60 KO and MYCN-amp cells show an increase of A-to-I editing, underscoring the complex and dynamic regulation of A-to-I editing.

Here, we compare these global trends detected by LoDEI with the results of the Alu Editing Index (AEI). The AEI reports a single value for each sample that summarizes the global amount of A-to-I editing of a sample and thus does not keep any positional information of the A-to-I editing events. We calculate the AEI for each sample and take the mean of all AEI values of a set as a representative of the A-to-I editing (Table 1, Supplementary Fig. 3).

The means of the AEI values are smaller for ADAR1 KD cells and sncRNA7SL OE cells compared to their control counterparts and support the global characteristics of a decrease of the A-to-I editing as detected by LoDEI (Table 1 rows 1 and 2). Both tools report a strong reduction in A-to-I editing in the ADAR1 KD cells and a small reduction in the sncRNA7SL OE cells. Similarly, both tools calculate an increase in A-to-I editing in RO60 KO and MYCN-amp cells (Table 1 rows 3 and 4).

In the ADAR1 KD dataset, the observed difference of −1.17 of the average AEI values is much stronger compared to the change of −0.06 in the sncRNA7SL OE dataset (*p* values < 0.01). Consistently, LoDEI

detects more windows with decreased A-to-I editing in the ADAR1 KD dataset (1624 windows with *q* ≤ 0.1, Table 1) compared to the sncRNA7SL OE dataset (64 windows with *q* ≤ 0.1).

In contrast, the AEI differences of 0.09 and 0.18 indicate an increase—though not statistically significant—of the A-to-I editing for the RO60 KO and MYCN-amp cells. Likewise, LoDEI detects 114 and 217 differential A-to-I windows with increased ($\delta^{A \rightarrow G} > 0$) A-to-I editing and *q* values ≤ 0.1 in these datasets.

In summary, the global trends detected by LoDEI are supported by the AEI. The number of found windows by LoDEI and the differences in the AEI values show a Pearson correlation of 0.99. Both tools identify an overall decrease of A-to-I editing in ADAR1 KD and sncRNA7SL OE cells and an increase in RO60 KO and MYCN-amp cells.

## LoDEI detects more A-to-I editing at the same FDR compared to alternative methods while maintaining signals found by single-site approaches

To the best of our knowledge, no window-based approach, including a statistical framework for differential A-to-I editing detection, exists besides LoDEI. Due to the lack of a window-based competitor, we used the single-site detections offered by REDIT and JACUSA2 to assess and compare the differential A-to-I editing detection performance. To obtain *q* values from REDIT and JACUSA2 for a direct comparison with results from LoDEI, we used a similar procedure as utilized in LoDEI. Therefore, REDIT and JACUSA2 were applied on G/A mismatches to approximate the number of false positives to finally calculate *q* values for detected differential A-to-I editing.

Overall, LoDEI detects more A-to-I editing sites at any *q* value threshold for all datasets compared to REDIT and JACUSA2 (Fig. 3). REDIT and JACUSA2 only detect differential A-to-I editing in the ADAR1 KD dataset (Fig. 3a) and could not detect any differential A-to-I editing in the other three datasets at reasonable *q* values (Fig. 3b–d). LoDEI's detected windows can contain both single and clusters of A-to-I editing sites (Supplementary Figs. 4 and 14).

At a *q* value threshold of 0.05, LoDEI detects 1231 differentially edited non-overlapping windows containing 79% (959) of the 1219 single sites found by REDIT in the ADAR1 KD dataset, demonstrating that LoDEI detects the majority of single sites detected by REDIT. In contrast, 52% (639) out of the 1231 windows detected by LoDEI contain 1795 single sites that are exclusively found by LoDEI and missed by REDIT (Supplementary Figs. 5 and 6).

With 2276 found sites at the same *q* value of 0.05, JACUSA2 detects more A-to-I sites than REDIT, and less than LoDEI. Of those 2276 JACUSA2 sites, 78% (1784) are overlapping with the 1231 windows detected by LoDEI showing that the majority of JACUSA2 sites are detected by LoDEI (Supplementary Fig. 5). The 26% (322) of windows uniquely found by LoDEI correspond to 818 sites missed by JACUSA2.

Overall, more sites are found by LoDEI at the same FDR compared to REDIT and JACUSA2 in all datasets (Fig. 3). In experiments with strong A-to-I editing changes, LoDEI can detect differential A-to-I editing between single samples (Supplementary Figs. 15 and 16).

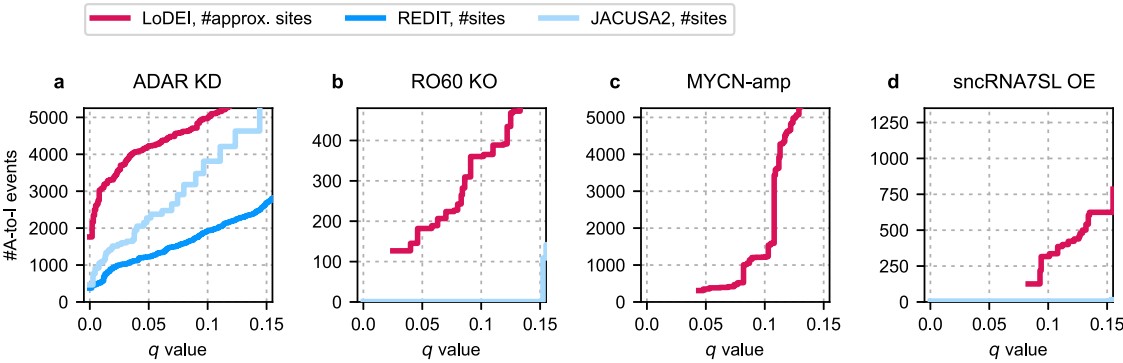

**Fig. 3 | Performance comparison of differential A-to-I site detection.** Shown are the number of detected differentially edited A-to-I sites as a function of the *q* value threshold for the ADAR1 KD (**a**), RO60 KO (**b**), MYCN-amp (**c**), and sncRNA7SL OE (**d**) datasets.

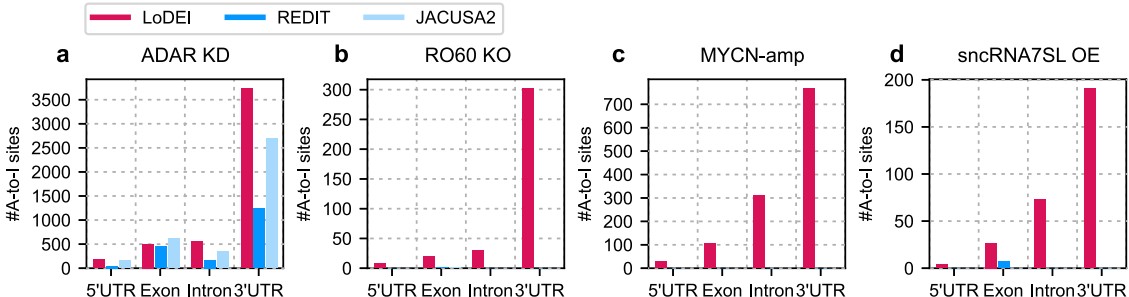

**Fig. 4 | Genomic locations of detected differentially edited A-to-sites.** The number of differentially edited A-to-I sites found by LoDEI (red), REDIT (blue), and JACUSA2 (light blue) at a *q* value threshold of 0.1 at different genomic locations are shown for the ADAR1 KD (**a**), RO60 KO (**b**), MYCN-amp (**c**), and sncRNA7SL OE (**d**) datasets.

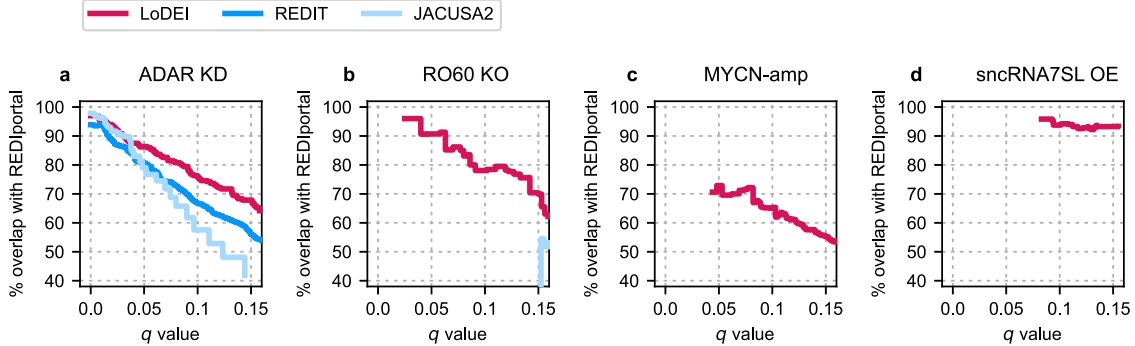

**Fig. 5 | Overlap of detected differentially edited A-to-I events with REDIportal A-to-I sites as a function of the *q* value.** The percent of overlap of detected differentially edited A-to-I events found by LoDEI (red), REDIT (blue), and JACUSA2 (light blue) with REDIportal A-to-I sites are shown for the ADAR1 KD (**a**), RO60 KO (**b**), MYCN-amp (**c**), and sncRNA7SL OE (**d**) datasets.

### LoDEI detects A-to-I editing at different genomic locations

In the ADAR1 KD datasets the majority of all detected differential A-to-I sites by LoDEI, REDIT, and JACUSA2 are located in 3'UTRs for all datasets (Fig. 4). However, whereas LoDEI detects A-to-I editing in all genomic regions of the RO60 KO, MYCN-amp and sncRNA7SL OE datasets, REDIT and JACUSA2 do not detect any editing, suggesting a high sensitivity for A-to-I editing by LoDEI. Most differential edited sites detected by LoDEI are located in the 3'UTR, followed by sites in introns, exons, and in 5'UTRs. This order of detected locations remains identical for all datasets, but the relative occurrences differ slightly between the datasets (compare Fig. 4a, b vs. Fig. 4c, d).

Taken together, we demonstrate that LoDEI detects differential A-to-I editing with higher sensitivity in all regions of mRNAs when compared to REDIT and JACUSA2.

### Differential A-to-I events detected by LoDEI overlap with known editing sites in the REDIportal

We tested whether A-to-I events detected by LoDEI are listed in the REDIportal database to provide further evidence for real differential editing events. As of this writing, the REDIportal consists of around 16 million A-to-I sites based on 9642 human RNAseq samples from 31 tissues of the GTEx project[23].

In general, for small *q* value thresholds, most detected A-to-I events by LoDEI overlap with editing sites listed in the REDIportal, and the percent of overlap decreases with increasing *q* value thresholds (Fig. 5). At a *q* value threshold of 0.05, 75.9%, 78.1%, 65.3%, and 93.8% of A-to-I events detected by LoDEI in the ADAR1 KD, RO60 KO, MYCN-amp, and sncRNA7SL OE datasets respectively, are overlapping with the REDIportal database. Of note, since REDIT and JACUSA2 did not detect differential A-to-I editing in all but the ADAR KD dataset, only

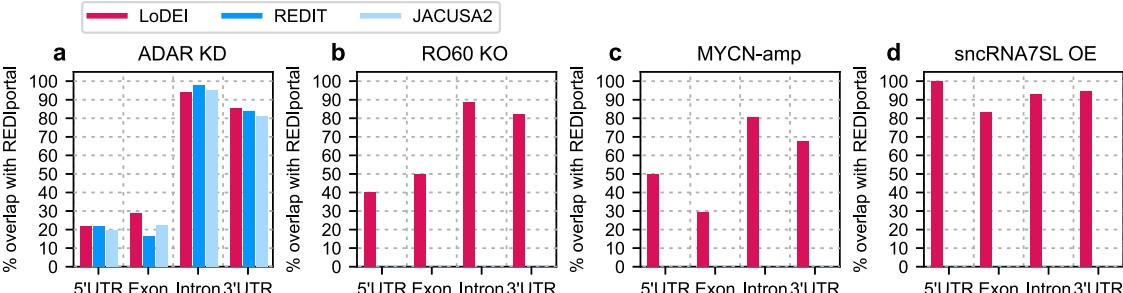

**Fig. 6 | Overlap of detected differentially edited A-to-I events with REDIportal A-to-I sites for different genomic locations.** The percent of overlap of detected differentially edited A-to-I events found by LoDEI (red), REDIT (blue) and JACUSA2 (light blue) at a $q$ value threshold of 0.1 at different genomic locations with REDIportal A-to-I sites are shown for the ADAR1 KD (**a**), RO60 KO (**b**), MYCN-amp (**c**), and sncRNA7SL OE (**d**) datasets.

minor or no overlap of the REDIT and JACUSA2 datasets with the REDIportal database was found, except for the ADAR KD dataset.

Next, we performed the REDIportal overlap analysis for the different genomic locations (Fig. 6). Again, only in the ADAR KD dataset a strong overlap with A-to-I sites available in the REDIportal database was found for all tools, however, in the RO60, MYCN, and sncRNA7SL datasets, with moderate changes in A-to-I editing (Fig. 2), only differential A-to-I editing sites detected by LoDEI are also found in REDIportal.

Taken together, the majority of A-to-I sites detected by LoDEI are also found in REDIportal. Even sites detected by LoDEI but not detected by REDIT or JACUSA2 were found in REDIportal, supporting the high sensitivity and robustness of LoDEI.

## Discussion

In this study, we present the *lo*cal *d*ifferential *e*diting *i*ndex (LoDEI) to detect differential A-to-I editing in two sets of RNA-seq samples by utilizing an empirical $q$ value calculation to separate signal from noise. LoDEI detects more differential A-to-I editing at the same FDR while maintaining the majority of editing sites found by currently available methods. Previously, two broad strategies existed to detect differences in A-to-I editing: the detection of single editing sites as suggested by REDIT and JACUSA2 and an overall comparison of whole samples as provided by the AEI. Since the editing characteristics of ADAR1 are a challenging scenario for any single-site detection software and the overall comparison provided by the AEI does not keep any positional information, LoDEI introduces a sliding window approach. A sliding window can handle ubiquitous editing and retains positional information compared to the global AEI. The choice of window size affects the resolution and number of the detected windows (Supplementary Figs. 7 and 11). Our analysis shows that LoDEI's empirical $q$ value calculation is applicable for A-to-I detection in RNA-seq data and outperforms current approaches.

We decided to use six different datasets to cover a broad range of protocols and experiments with different levels of difficulty for evaluating the general applicability and robustness of the tested programs. Since ADAR1 is known to be the primary protein for A-to-I editing and the reduction of editing upon knockdown of ADAR1 could be shown in multiple studies, we consider the ADAR1 KD dataset as the easiest of the four datasets[16,37]. As expected, LoDEI could detect significant differential A-to-I editing in the ADAR1 KD dataset and was the most sensitive approach detecting more differential A-to-I editing events compared to JACUSA2 and REDIT. Besides A-to-I sites, LoDEI also detected a small number of $T \rightarrow C$ mismatches that have recently been described as being caused by ADAR RNA editing on antisense RNAs overlapping with sense transcripts[41].

Since it was shown in multiple studies that the knockout of RO60 results in an increase of the A-to-I editing, but at a lesser extent as the decrease of editing as detected upon ADAR1 KD, we expected to find differential A-to-I editing in the RO60 KO dataset while considering this dataset to be more challenging due to the weaker effects of the RO60 knockout on A-to-I editing[24,37,38]. Interestingly, only LoDEI could detect significant differential editing in this dataset, whereas the other editing detection programs failed to identify significant changes.

Taken together, LoDEI's outcomes are in agreement with the previously published results that the ADAR1 KD reduces and that the RO60 KO increases A-to-I editing.

In contrast to all other datasets, the MYCN-amp dataset consists of RNA-seq data of different cell lines, and we show that A-to-I editing is positively regulated in MYCN-amp cells. To the best of our knowledge, this is the first report demonstrating the role of MYCN in the regulation of A-to-I editing. The AEI results support LoDEI's findings and show differences even within identical conditions (Supplementary Fig. 3). Hence, A-to-I editing is a highly dynamic and diverse process even in related cells, suggesting that A-to-I detection in heterogeneous cohorts is a challenging scenario successfully addressed by LoDEI. Besides A-to-I editing signals, LoDEI detected a small number of $G \rightarrow A$ differences. Other studies analyzing A-to-I editing data also made similar observations that significant non-$A \rightarrow G$ mismatches can occur in RNA-seq datasets[42].

The MYCN-amp dataset, as a representative of a dataset where different cell lines are compared, is challenging for any differential editing detection. Any software that utilizes observed mismatches from NGS data can be affected by SNPs. Note that in experiments where cell lines are compared against each other (e.g., the ADAR KD, RO60 KO, and sncRNA7SL datasets), SNPs do not affect LoDEI's $\delta^{A \rightarrow G}$ values, since SNPs are part of every sample in both sets and do not impact the difference between the sets (Eq. (3), Supplementary Fig. 17). However, in scenarios where the compared sets consist of different types of cells, SNPs can have an effect that is correctly reflected by the empirical $q$ value estimation. The non-$A \rightarrow G$ mismatches used for the empirical $q$ value estimation are affected in the same way and thus also contain a similar amount of SNPs. As a consequence, empirical $q$ values will start to increase and correctly inform the user about the expected number of false positives in the analysis. In cases where SNPs start affecting the data heavily (i.e., haven't been removed), the empirical $q$ values correctly reflect this situation in the provided results and indicate that SNPs should be treated explicitly upfront in the analysis pipeline.

sncRNA7SL was tested as a potential modulator for A-to-I editing because this 32 nts long small RNA has high complementarity to specific reverse Alu elements and might be able to disturb the folding of Alu and inverted repeated Alu (IRAlu) into long double-stranded RNAs. In the sncRNA7SL OE dataset, LoDEI detects a decrease in A-to-I editing after sncRNA7SL transfection, supporting this assumption.

Interestingly, most differential signals are detected for $G \to C$ mismatches, an observation not being made in any other analyzed dataset in this study. How the observed $G \to C$ transversion is induced upon sncRNA7SL OE is currently unknown.

LoDEI is the only program that detected differential A-to-I editing in all tested datasets, demonstrating general applicability and robustness across various experimental settings and in different organisms. The correlation of the general trends of the AEI and LoDEI supports LoDEI's findings. The genomic locations of the detected differential A-to-I editing by LoDEI further support the found windows and our results are in agreement with the current literature that A-to-I editing primarily takes place in 3'UTRs. The large overlap of detected differential A-to-I events with editing sites listed in the REDIportal further supports our results. Of note, detected differential A-to-I editing not overlapping with REDIportal data does not indicate false positive detection as exemplified (Supplementary Fig. 10). Since the data offered by the REDIportal does not concentrate on differentially edited sites, the comparison might be limited in that way. From a theoretical perspective, differentially edited sites should be a subset of generally edited A-to-I sites.

For the empirical $q$ value calculation, we decided to use all non-$A \to G$-mismatches, even knowing that some non-$A \to G$ mismatches like $C \to T$ mismatches can be valid editing caused by other proteins than ADAR. With keeping all non-$A \to G$ based $q$ value estimates, the final $q$ values for $A \to G$ mismatches will tend to be rather higher than lower $q$ values and thus are conservative estimates.

In conclusion, LoDEI's empirical approach detects differential A-to-I editing in RNA-seq data in a robust fashion across different experimental protocols and offers a more sensitive method to find differential A-to-I editing even in challenging conditions and any organism. We are convinced that the high sensitivity and deep positional resolution provided by LoDEI will advance our understanding of the dynamics and regulation of A-to-I editing in basic research. In addition, LoDEI should be of high value for addressing clinical questions such as the role of differential A-to-I editing in disease, the detection of off-target effects in drug development, and the use of designed base editors.

## Methods

### Cell culture growth and splitting
Cell culture media and their ingredients were stored at 4 °C or −20 °C and warmed up to 37 °C in a water bath before use. All work was performed under sterile conditions, and the neuroblastoma Kelly cell line (Leibniz Institute DSMZ) was cultivated in tissue-culture-treated culture dishes (Sarstedt), in a 5% $CO_2$ atmosphere and at an environmental temperature of 37 °C. Routinely, Kelly cells were tested for mycoplasma contamination by applying the PCR Mycoplasma Test Kit I/C from PromoCell. Kelly cells were cultured in RPMI 1640 (Thermo Fisher) containing 10% FBS (Thermo Fisher), 1% Penicillin–Streptomycin (Sigma-Aldrich), and 2 mM L-Glutamine (Thermo Fisher).

### Reverse transient transfection
The transfection was performed with lipofectamine RNAiMAX (Thermo Fisher). For this, 6 µl of RNAiMAX was diluted in 125 µl OptiMEM (Thermo Fisher). Seven microliters of RNA oligonucleotides (sncRNA7SL or sncRNACtrl; stock = 10 µM; Horizon Discovery) were also diluted in 125 µl OptiMEM (dilution 1:22) in OptiMEM. The two reactions were mixed (ratio 1:1) by pipetting up and down, incubated for 5 min at room temperature, and 250 µl was distributed in each well of a six-well plate (Omnilab). In parallel Kelly cells grown to 75% confluence were detached and 1.0 million cells were resuspended in 2 ml medium. The 2 ml cell suspension was added to the transfection

reaction of one well, leading to a final concentration of 25 nM RNA oligonucleotides per well.

```
RNA oligonucleotides used:
RNA modifications:
* = Phosphorothioate bond,
m = 2'-O-methylation of RNA bases,
sncRNA7SL (32 nts):
5'-P-rA*rG*rC*rC*rU*rG*rA*rG*rC*rA*rA*rC*rA*rU*r
A*rG*rC*rG*rA*rG
*rA*rC*rC*rC*rC*rG*rU*rC*rU*rC*rU*mA-3'
sncRNActrl (32 nts):
5'-P-rA*rU*rC*rU*rC*rU*rG*rC*rC*rC*rC*rA*r
G*rA*rG*rC*rG*rA*rU*rA
*rC*rA*rA*rC*rG*rA*rG*rU*rC*rC*rG*mA-3'
```

### RNA purification and total RNAseq
48 h after transfection, the cells were harvested by trypsinization, pelleted (300×$g$; 4 °C, 5 min), the supernatant removed, and the pellets were resuspended in PBS (Thermo Fisher) for washing. After another centrifugation (300×$g$; 4 °C, 5 min), total RNA was prepared from the cell pellet using the RNAeasy mini kit (Qiagen) with an integrated DNAse digestion step according to the manufacturer's protocol. The RNA was eluted from the column with RNase-free water and stored at −20 °C. 250 ng of total RNA was used for RNA seq analysis. RNA quality control library preparation and RNAseq were performed at the Genomics Core Facility "KFB-Center of Excellence for Fluorescent Bioanalytics" (University of Regensburg, Germany). Library preparation and RNAseq were carried out as described in the Illumina "Stranded mRNA Prep Ligation" Reference Guide, the Illumina NextSeq 2000 Sequencing System Guide (Illumina, Inc., San Diego, CA, USA), and the KAPA Library Quantification Kit-Illumina/ABI Prism (Roche Sequencing Solutions, Inc., Pleasanton, CA, USA). Equimolar amounts of each library were sequenced on an Illumina NextSeq 2000 instrument controlled by the NextSeq 2000 Control Software (NCS), using one 100 cycles P2 Flow Cell with the dual index, single-read (SR) run parameters. Image analysis and base calling were done by Real Time Analysis Software (RTA) v3.9.25. The resulting. cbcl files were converted into. fastq files with the bcl2fastq v2.20 software.

### Datasets and annotations
The human genome (GRCh38.p14) and corresponding annotations (release 44) were downloaded from GENCODE and used for all analysis[43]. The ADAR1 knockdown dataset was previously published by Bahn et al. and is accessible via Gene Expression Omnibus (GEO) under the accession number GSE28040[16]. The RO60 knockout dataset has the GEO accession number GSE72501, and the MYCN-amp dataset is available via GSE145075[17,39]. The sncRNA7SL OE dataset was generated for this publication and is available via the GEO accession number GSE263010. The *C. elegans* dataset is available via the GEO accession number GSE83133[44].

### RNA-seq analysis
Raw sequencing data was downloaded from the NCBI Sequence Read Archive via fastq-dump. Snakemake was used for analysis workflows for all datasets[45]. If not stated otherwise, programs were run with default parameters.

Fastq files were quality filtered using cutadapt with the parameters –minimum–length 25 –cut 5 –q 20[46]. To reduce the chance of false positives, we recommend proper quality filtering of the sequencing reads prior to differential A-to-I detection. Filtered reads were aligned to the human reference genome using STAR with default parameters[47]. Sorted BAM files served as input for the A-to-I editing

analysis by LoDEI, REDIT, JACUSA2, and the AEI index[27–29]. LoDEI, REDIT, and JACUSA2 were used to detect differential A-to-I editing in protein-coding genes. The AEI was run with the default annotation provided in the AEI container image (see below).

If applicable, a minimum coverage of 10 was used for all A-to-I editing detection software. LoDEI was run with default parameters except for the MYCN-amp dataset, where the `-rm_snps` flag was used to activate SNP removal. As an ad-hoc filter to remove potential SNPs, a position is excluded from the calculation, if a single position in any of the samples of a set shows a mismatch frequency ≥80%.

As recommended by the AEI documentation, a Docker container was built from the provided Dockerfile (https://github.com/a2iEditing/RNAEditingIndexer), and the analysis was performed according to the authors' documentation on GitHub.

REDIT was downloaded from the corresponding GitHub repository (https://github.com/gxiaolab/REDITs) and p-value calculation was performed according to the authors' documentation.

Q-values for REDIT and JACUSA2 were computed using a similar empirical approach as used in LoDEI. REDIT and JACUSA2 were used to calculate results for G/A mismatches that were used to approximate the number of false positives to calculate q values for detected A/G mismatches.

The RNA-seq data for the ADAR1 KD and RO60 KO datasets is unstranded. To assign the location of A-to-I sites unambiguously only those sites and windows reported by all compared tools were used that overlap uniquely to gene annotations.

### Reporting summary
Further information on research design is available in the Nature Portfolio Reporting Summary linked to this article.

## Data availability
The ADAR1 knockdown dataset was previously published by Bahn et al. and is accessible via Gene Expression Omnibus (GEO) under the accession number GSE28040. The RO60 knockout dataset has the GEO accession number GSE72501 and MYCN-amp dataset is available via GSE145075. The sncRNA7SL OE dataset is available via the GEO accession number GSE263010. The *C. elegans* data is available via the GEO accession number GSE83133. For the reproducibility of the paper, the complete software stack used to generate the analysis shown in this paper is available as a Podman/Docker image together with additional information, all result files and a manual at Zenodo https://doi.org/10.5281/zenodo.12748069[48].

## Code availability
LoDEI is GPLv3-licensed free software. The source code, as well as a detailed manual, is available at GitHub (https://github.com/rna-editing1/lodei), and a corresponding Podman/Docker image of the latest version is available at DockerHub (https://hub.docker.com/r/lodei/lodei)[49]. A small test dataset for LoDEI is available via Zenodo at https://doi.org/10.5281/zenodo.12748864[50].

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

## Acknowledgements

We thank Marie Matthes (Universitätsklinikum Regensburg, Germany) and Maria Bösl (University of Regensburg, Germany) for help and input at the beginning of the A-to-I editing project. RNA quality control library preparation and RNAseq were performed at the Genomics Core Facility KFB-Center of Excellence for Fluorescent Bioanalytics (University of Regensburg, Germany). This work was funded by the German Research Association Deutsche Forschungsgemeinschaft (DFG) [SFB 960/3, project B14], which is gratefully acknowledged.

## Author contributions

P.T. designed the differential editing analysis, implemented the software, carried out the analysis, interpreted results, and wrote the manuscript; M.S. performed the cell culture experiments, sample preparation, and analyzed RNA seq data; U.S. contributed to the RNA seq data analysis; S.C. contributed to the interpretation of the results; G.S. conceived the original idea, contributed to the software development, and the interpretation of the results; T.H. conceived the original idea, contributed to the software development and design of the differential editing analysis, manually evaluated the results, wrote the manuscript, and supervised the project. All authors discussed the results, contributed to the final manuscript, accepted responsibility for the entire content of this submitted manuscript, and approved the submission.

## Funding

## Competing interests

The authors declare no competing interests.
