## [Peer Review File · Nature Communications]

LoDEI: a robust and sensitive tool to detect transcriptome-wide differential A-to-I editing in RNA-seq dataREVIEWER COMMENTS

Reviewer #1 (Remarks to the Author):

In this study, Torkler et al. introduce a novel bioinformatics tool, the local differential editing index (LoDEI), designed for the detection of transcriptome-wide differential A-to-I editing events. Through a comparison of LoDEI with three existing methods (e.g., JACUSA2, REDIT) using publicly available RNA-seq datasets (ADAR knockdown, Ro60 knockout, MycN-amplified, and sncRNA7SL overexpression), the authors concluded that LoDEI offers superior sensitivity and accuracy in identifying global transcriptome-wide RNA editing changes, while maintaining precise positional resolution in RNA editing regulation. Overall, the authors have introduced a novel and promising method that could fill a significant research gap in the study of RNA editing changes within the field of RNA biology. However, this manuscript lacks the depth and substantial evidence necessary to fully support the claims regarding the improved sensitivity and accuracy of the proposed method.

Specific points:

1. The authors showed that the LoDEI method is more sensitive and detects more A-to-I editing sites than REDIT and JACUSA2 in all four datasets (Results section 2.4 lines 461-489). However, the authors do not provide evidence that these newly identified editing sites are true positive differential editing sites. The authors should perform experimental validation of their findings using a low throughput method to demonstrate that the method can detect true editing events missed by other methods.
2. Although the use of real RNA-sequencing data is useful for biological interpretation, the accuracy of LoDEI in detecting differential editing events cannot be properly determined as there are no established ground truths in the datasets. In their performance comparison of differential A-to-I site detection methods, it will be more useful if the authors can use datasets that have well-established ground truths in differential editing sites and differential editing effect sizes. In the lack of such datasets, the authors should use simulated data with specific editing levels and editing changes to evaluate the accuracy of the proposed method in comparison to other current methods. The simulation datasets should also include varying levels of signal-to-noise ratios to demonstrate the robustness of LoDEI in noisy data.
3. As there is a lack of any meaningful biological interpretation of the new results, such as how the identified novel editing sites might be implicated in disease, the biological significance of the results is unclear.

Additional comments

1. The LoDEI method uses on a sliding window approach for q-value calculation. This approach partly relies on the assumption that A-to-I RNA editing regulation are spatially correlated (i.e., editing regulation of sites spatially close together in terms of genomic coordinates are correlated). If this assumption does not hold true, there might be a higher false discovery rate than presented. The authors should provide evidence of this or comment in more detail.
2. In relation to the sliding window approach, the software provides options for specifying window size and step size for wEI and q-value calculation (lodei find). It would be useful if the authors can describe how the window and step size might impact the validity and robustness of the results, and provide recommendations for the use of non-default window

and step sizes for different datasets. In addition, the authors should perform additional analyses using different combinations of window and step sizes (e.g., -w 100 -s 50, -w 200 -s 100, -w 100 -s 25) to show that the presented results are not variable when using different parameters.

3. The authors proposed that LoDEI can detect differential editing between two sets of samples. The use of multiple biological replicates can tremendously improve the reliability and reproducibility of analysis results as it allows the estimation of signal-to-noise ratio and variability within biological groups. In the method description of LoDEI, A-to-I editing signals of all samples of a set for any given window are simply estimated using “the mean of the editing signals of all samples of a set” (line 235). In the following step, the change in editing between two sets are calculated by taking the difference between the mean of editing signals. This approach does not seem to consider reproducibility of editing changes across replicates and could be biased by outliers. The representation of a q-value for each wEI in this context is also somewhat misleading as it might be interpreted as a measure of statistical significance of the difference between two sets of samples. The authors should implement some improvements in their software to address this, such as including a statistical test (e.g., Wilcoxon rank test) between two sets of editing signals. Otherwise, the authors should provide some comments and clarification on this issue.

4. The use of a global q-value estimation is likely heavily influenced by the overall signal-to-noise ratio or sequencing quality of the RNA-seq dataset. This might decrease the method’s sensitivity and accuracy of detecting editing changes with small effect sizes even if they are consistent. The authors should give evidence that these are true negatives or provide some comments on this.

5. In the Results section 2.1 line 269-273, the statement that “we assume increasing q values for strong values of δ as counterintuitive and avoid varying q value estimates by not allowing those estimates to increase again” is very vague and unsupported. The authors should provide some statistical/mathematical evidence or clarification for this step.

6. The authors should provide more biological interpretation of the new results in this paper. With the identified novel differential editing sites and novel regulators of A-to-I editing, the authors should investigate and elaborate on their roles and functions in biological functions and how their dysregulation might be implicated in diseases such as cancer or autoimmune disease. The authors should also perform additional differential RNA editing analyses in disease model or patient sample RNA-seq data to validate their novel results to more clearly demonstrate the biological significance of their new findings.

Reviewer #1 (Remarks on code availability):

The README on the GitHub repository is written clearly with instructions to run the software. I tested the software installation using conda and pip and was able to successfully run the test data provided. However, the authors should include more details of the results in the Output section, such as what columns are expected in the output files and what they mean, how merged scores were calculated etc.

Reviewer #2 (Remarks to the Author):

The authors present a new method for testing for differential A-to-I editing from RNA sequencing data. The main novelty on the approach lies in the use of a sliding window, which can potentially increase power by capturing editing sites in groups instead of considering them singularly (also biologically relevant given ADAR's indiscriminate editing behaviour in double stranded regions). They further incorporate non-parametric testing of differential editing by considering a null-distribution of change observed across non A-to-G edit types, test on 4 datasets and find that their method generally detects more differentially edited sites compared to JACUSA2 and REDIT.

Overall, developing novel approaches (sliding window/ non-parametrics) in this context is useful given there currently lacks methods for differential A-to-I editing. However, to be adopted by the community LoDEI requires a lot more testing with regards to the usefulness/robustness of the approach, see comments below.

> By taking the difference of two averages instead of the average of the differences in the window you are removing the paired information of the individual sites - could this result in a loss of statistical power? Some discussion of this and more details regarding the reasoning behind the methodological decision making would be useful. Also, by summing sites, the inherent variability in editing across the replicate samples is lost - some discussion of this aspect is required.

> The 'sliding window approach' is not well explained in the methods/text - how exactly are the windows calculated, are they non-overlapping (like depicted in Figure 1B)? What is the size of the window (the chosen 'w') for the various studies and how did the authors come forward to the most appropriate 'w' for each? Reading the instructions on the Github page actually did help figure out the defaults, but more needs to be said in the text.

> What is the overall distribution of the numbers of sites in the windows transcriptome wide (are there many containing singletons or does it generally capture clusters)? Can different 'information' be obtained by using different window sizes in the same comparison?

> It is claimed that LoDEI is applicable for comparisons between single sample, but I do not see any convincing demonstrating/testing of this idea by comparing the results for 1-2-3 samples from the test datasets.

> Figure 2. The y scale can vary a lot, down to -300 in some cases. I guess it depends on the expression at the individual site, but wouldn't some kind of normalised score make sense?

> Use of the threshold $q < 0.1$ - would be helpful with more consideration/discussion about how to tune the threshold for capturing the most biological relevant changes.

> Figure 3: For the ADAR KO data, one would specifically expect many of the significantly reduced sites upon KO to be supported by REDportal - what is the precision/recall of the LoDEI method with respect to REDportal (which I realise is not perfect but still highly relevant), and compared to the other other methods? Also, some discussion of how JACUSA/REDIT actually work and differ in terms of their methods, and how they were applied to be optimally comparable with LoDEI is important.

> Figure 4 ditto - how many of the sites found in b-d are editing sites supported by REDportal? And in 4a) how many sites/regions are shared by the methods, and are the shared ones driven by larger changes in editing proportion?

> Much of the data used in the paper likely has very large shifts in ADAR explaining many of the changes/global shifts in editing. Showing extra supporting figures to show the direction of ADAR change would be useful in the context of understanding the results.

> Since C>U can be valid editing (mediated by APOBEC) it seems strange to include it in the list of possible background noise (though I acknowledge that the median strategy might filter this, but more discussion would be useful).

> Treatment of SNPS appears somewhat downplayed in the focus of the article but one can imagine the existence of SNPs compared to the reference would really inflate the background null, including heterozygous SNPs which are even less obvious to see. More description of the methods and discussion about the steps taken to deal with SNPs should be included.

Reviewer #2 (Remarks on code availability):

Note: I have read the GitHub front page which appears well explained but do not have available time to run the code myself.

Reviewer #3 (Remarks to the Author):

Reviewer #4 (Remarks to the Author):

In this article, Torkler et al. presented a novel method to more sensitively detect differential A-to-I editing in RNA-seq datasets based on a sliding-window approach coupled with an empirical q-value calculation. The authors called this method the “local differential editing index” (LoDEI). The LoDEI approach was compared with results from the site-specific differential A-to-I detection tools, such as REDIT and JACUSA2, as well as with results of the global A-to-I detection as provided by the Alu Editing Index (AEI). Although the results presented by the authors are interesting, I have several comments that should be addressed before being considered for publication. My comments are detailed below.

Major comments:

- One of the major points of discussion is the authors' choice of the window length, denoted as 'w'. It needs to be made clear how this value is determined. Is it a user-defined

parameter? The choice of the window length is a crucial aspect of the LoDEI calculation, and it would be beneficial for the authors to explain their rationale. Additionally, it would be insightful to see how varying the window length impacts the number of significant differences.

- Following my comments above, regarding the length of the window, is there any difference between samples from cell lines and tissue samples? Also, as the RNA editing phenomenon depends on RNA structures, in which repetitive elements play an essential role, how does the length of the window impact different species, such as mice and drosophila, compared to humans? Finally, what is the size of the window chosen for the analyses performed in the study, and what is the justification for such a choice?
- At the beginning of Section 2.4, lines 451 and 452, the authors stated that besides LoDEI, no window-based approach exists, including a statistical framework for differential A-to-I editing detection. However, I would recommend considering these two previously published tools, such as RNAEditor and FLARE (PMIDs: 27694136, 37784060), which use a "window-" or "cluster-"based approach and see if there are any differences or similarities with the LoDEI tool. The authors could also assess if these tools are suitable for detecting RNA editing differences between different conditions. If not, at least they could compare them with LoDEI regarding detecting A-to-I editing sites.
- The authors choose four different datasets to compare LoDEI with other methods, such as REDIT and JACUSA2. These are suitable datasets to make these comparisons; however, they are limited as they are derived from human cell lines. As REDIT and JACUSA2 tools and others were employed in several other sample data, I would suggest having a more comprehensive comparison with human tissue samples, for example, data from TCGA, TARGET, or GTEx (even choosing a tissue type as a case study), or different species (e.g., mouse, drosophila).
- The authors should put more information about the mapping (including the parameters used for the STAR tool) and RNA modification annotation steps. For example, they specified that the filtering of SNPs was only employed for the MycN-amp dataset. Such a filter is an essential step in the RNA editing characterization, and I recommend including it as a mandatory step in the workflow. Moreover, several of the RNA editing detection tools, to reduce the number of false positives in the modification events characterization, employ Bayes models or Binomial tests based on the quality base of nucleotide with editing event (information present in the FASTQ files) and filtering those RNA editing events with low score or not significant. I suggest the authors include this aspect in the modification event characterization workflow to reduce the number of false positives. Several other filters used by other tools can help reduce the false positives in RNA editing detection, and it would be helpful to at least discuss some of them in the text and adopt them in the workflow.

Minor comments:

- The authors should try to explain why they used these datasets and give an idea of the expected outcomes from the comparisons in the first paragraph (lines 279 to 283). This was done in Section 2.2.1, but it would be better to anticipate it so the reader could easily follow the flow of the results presented in Figure 2.
- In lines 671 and 672, the authors stated, "a position is excluded from the calculation, if a single position in any of the samples of a set shows a mismatch frequency 80%." However, this needs a reference or at least justification.

We thank all reviewers very much for their constructive comments and critiques. We addressed most of the reviewer's remarks experimentally and addressed others by explaining certain aspects in more detail. In sum, we added two new figures to the main manuscript and generated a new supplementary information document consisting of 15 new figures and one new table. We are convinced the manuscript is significantly improved due to the reviewer's comments.

As you will see below, we numbered all our responses to the reviewers' remarks and addressed them point by point. In some cases, similar comments were given by different reviewers. If possible, we bundled the response as indicated. Additional information, new data, and figures are included in the supplementary information, as indicated. Changes in the main body of the manuscript are shown in this rebuttal letter and are highlighted in the manuscript itself.

Reviewer #1 (Remarks to the Author):

In this study, Torkler et al. introduce a novel bioinformatics tool, the local differential editing index (LoDEI), designed for the detection of transcriptome-wide differential A-to-I editing events. Through a comparison of LoDEI with three existing methods (e.g., JACUSA2, REDIT) using publicly available RNA-seq datasets (ADAR knockdown, Ro60 knockout, MycN-amplified, and sncRNA7SL overexpression), the authors concluded that LoDEI offers superior sensitivity and accuracy in identifying global transcriptome-wide RNA editing changes, while maintaining precise positional resolution in RNA editing regulation. Overall, the authors have introduced a novel and promising method that could fill a significant research gap in the study of RNA editing changes within the field of RNA biology. However, this manuscript lacks the depth and substantial evidence necessary to fully support the claims regarding the improved sensitivity and accuracy of the proposed method.

Specific points:

1. The authors showed that the LoDEI method is more sensitive and detects more A-to-I editing sites than REDIT and JACUSA2 in all four datasets (Results section 2.4 lines 461-489). However, the authors do not provide evidence that these newly identified editing sites are true positive differential editing sites. The authors should perform experimental validation of their findings using a low throughput method to demonstrate that the method can detect true editing events missed by other methods.

1.1 We thank the reviewer for this important comment, which is also related to comment 1.2 and to reviewer #2 and our responses 2.7 and 2.8. First, to collect evidence and support the presented results, we followed the comment of reviewer #2, who suggested comparing the detected differential A-to-I sites with A-to-I sites stored in the REDlportal database (see also answers 2.7 and 2.8). We included an analysis comparing the detected sites with the REDlportal database to the main manuscript. Overall, the percent of overlap of differential A-to-I editing detected by LoDEI with the REDlportal database is >80% for q values < 0.1 except for the MycN-amp dataset (new Fig. 5), further supporting that the detected differential A-to-I sites are very likely to be

true editing sites. In addition, we rearranged the explanation for the different datasets (see 1.2).

Secondly, we selected four LoDEI windows and verified the identified differential A-to-I editing sites using the IGV genome browser. As shown, LoDEI detects A-to-I editing sites in RNA Seq data sets (Supplementary Fig. 1, 2 and new Supplementary Fig. 9 and 10).

Changes in the revised manuscript:
See answers 1.2, 2.7 and 2.8.

We added two new figures and legends to the main manuscript and two new figures and legends to the supplementary information:

Fig. 5

Fig. 6

Supplementary Figure 9

Supplementary Figure 10

2. Although the use of real RNA-sequencing data is useful for biological interpretation, the accuracy of LoDEI in detecting differential editing events cannot be properly determined as there are no established ground truths in the datasets. In their performance comparison of differential A-to-I site detection methods, it will be more useful if the authors can use datasets that have well-established ground truths in differential editing sites and differential editing effect sizes. In the lack of such datasets, the authors should use simulated data with specific editing levels and editing changes to evaluate the accuracy of the proposed method in comparison to other current methods. The simulation datasets should also include varying levels of signal-to-noise ratios to demonstrate the robustness of LoDEI in noisy data.

1.2 We thank the reviewer for this comment and apologize for not providing more evidence to support our claim that the detected differential A-to-I sites are likely to be true. Comment 1.2 is also related to comment 1.1 and to reviewer #2 comments 2.7 and 2.8.

First, we would like to highlight the differences between the chosen datasets. We decided to start the validation of LoDEI with the ADAR KD and Ro60 KO datasets since a change in the A-to-I editing signal is known for both datasets. ADAR is the enzyme catalyzing the adenosine to inosine deamination and an ADAR KD is known to cause a strong drop in A-to-I editing events. As expected and shown by others, the ADAR KD causes strong differences in A-to-I editing concomitant with strong biological effects [9, 10, 5]. Applying LoDEI to the ADAR KD dataset confirms changes in the editing signals (see Fig. 2 a vs Fig. 2 b, Supplementary Fig. 12 a vs. b and d vs. e). Likewise, as written in the manuscript, Ro60 KO yields an increase of A-to-I editing that could also be identified by using LoDEI. Notably, strong editing signal changes are observed exclusively for A/G mismatches but not for other mismatches.

In the ADAR KD and Ro60 KO datasets, the primary question is how many sites can be detected at what fraction of false positives and to compare the detection capability of LoDEI to all tested programs. Given that the ADAR knockdown and Ro60 KO causes the A-to-I editing, these signal changes are true changes in A-to-I editing. Therefore, we considered the ADAR KD and Ro60 datasets as the ground truth. We apologize that this explanation wasn't clear in the first manuscript. Reviewer #4 suggested (see answer 4.7) to reorder the paragraphs explaining the choice of the datasets to avoid such misunderstandings. We followed the suggestion of reviewer #4 and reordered the explanation of the datasets.

Changes in the revised manuscript:

We moved dataset explanations from the subsections 2.2.1 to section 2.2 to increase the readability of the manuscript.

The line 283 in the main manuscript

“Two of the four datasets are known to show differential A-to-I editing.”

changed to (lines 300-314)

“The first two of the four datasets are known to show differential A-to-I editing and are used to show the general applicability of LoDEI's approach. All datasets show a significant change in the gene expression in at least one of the genes of the ADAR family (Supplementary Table 2).

The first analyzed RNA-seq dataset consisting of two samples per set is known to show a strong reduction of A-to-I editing upon siRNA-induced ADAR1 knockdown (KD) in the glioblastoma cell line U87MG when compared to a control group [16]. Other RNA-binding proteins are known to regulate A-to-I editing besides ADAR, and several publications could show an increase in A-to-I editing upon the reduction of Ro60 (TROVE2) [24, 37, 38]. Hence, we used the RNA-seq dataset consisting of two control and three knockout samples derived from the Ro60 knockout (KO) Lymphoblastoid cell line GM12878 as the second dataset [17].

After evaluating the general applicability of LoDEI on the ADAR KD and Ro60 KO datasets, we applied LoDEI to novel datasets to search for differential A-to-I editing.”

In respect to the suggestion to develop an A-to-I simulation tool, we have to argue that the major drawback of simulated data sets is that they are not based on real-life situations like occurring in cell-based assays or even patients. Therefore, and as argued above we consider the ADAR KD and Ro60 KO data sets as the ground truth and RNA-seq data as the best source of A-to-I editing information.

Although, we agree that systems biology approaches can be powerful in simulating even biological systems, to develop a sophisticated A-to-I editing simulation tool integrating just a few basic regulatory mechanisms such as regulation of ADAR expression, tertiary

structures of RNA molecules, or IFN signaling, is out of the scope of this project aiming to develop a novel tool to detect differential A-to-I editing. Moreover, considering a simple A-to-I editing simulations approach, based on running LoDEI on in silico-designed models of more or less edited substrates, has many limitations when translating those findings to real cellular conditions as reflected by the four data sets studied herein.

Considering these thoughts and the high degree of overlap with sites listed in the REDportal database, we do not see an advantage of a simulation approach aiming to simulate cellular processes such as the highly dynamic A-to-I editing.

3. As there is a lack of any meaningful biological interpretation of the new results, such as how the identified novel editing sites might be implicated in disease, the biological significance of the results is unclear.

1.3 We thank the reviewer for giving us the chance to present more biological interpretations of the results. As pointed out in the introduction of the manuscript (lines 65 to 81, 134-141), the impact of dsRNA and the extent of A-to-I editing in these regions is of great interest for immunotherapies. In addition, some reports identified single A-to-I editing sites, which affect the expression and function of specific proteins forming membrane channels. These published data show a clear biological meaning of A-to-I editing. Therefore, any novel approach improving the identification of changes in A-to-I editing is beneficial for basic research and applied science.

To demonstrate how the identified novel editing sites might be implicated in disease, we provide two new examples (Supplementary Fig. 9 and 10) of A-to-I sites that potentially cause a point mutation in an open reading frame and a potential impact on alternative splicing.

In the new supplemental Figure 9 we show the coordinates of a LoDEI window (Ro60 KO data set) localized in the open reading frame of the signal recognition particle 9 (SRP 9) gene. Three A-to-I editing sites are located in the window identified by LoDEI and confirmed by the screenshot of the IGV genome browser window (indicated by arrows, (Supplementary Fig. 9). Considering that inosine residues are read as guanosine by cellular machines, A-to-I editing can, as example, result in mutation in the open reading frame and to mutations at splice sites. The first two sites lead to a potential silent mutation. However, the third site changes from a serine codon (AGC) to a glycine codon (GGC) likely caused by A-to-I editing. In this event, about 30% of the SRP9 protein would be an SRP9 isoform which cannot be phosphorylated anymore at serine 75. Furthermore, this A-to-I editing event was found in Ro60 KO cell lines, suggesting that fluctuation in Ro60 level could affect the regulation of SRP9 by phosphorylation.

In the second example, a LoDEI window from the MycN-amp data set, located in the open reading frame of the CHAF1A (Chromatin Assembly Factor 1 Subunit A) gene contains a 3' splice acceptor site that is affected by A-to-I editing. A change of this site could affect the splicing of the CHAF1A mRNA and further suggest that the increase in

A-to-I editing associated with MycN-amplification, as shown herein, could have an impact on CHAF1A expression (Supplementary Fig. 10).

Further analysis of the impact of differential A-to-I editing events on the cell is out of the scope of this manuscript. This publication aims to introduce LoDEI and demonstrate LoDEI's capability to identify novel and potentially biologically relevant differentially edited A-to-I sites.

Changes in the revised manuscript:

We added the following paragraphs to the main manuscript:

Lines 344-350:

“To demonstrate potential novel discoveries facilitated by LoDEI's results, we selected a LoDEI window with differential A-to-I editing in the open reading frame of the SRP9 gene in the Ro60 dataset. The analysis of the window in the IGV genome browser revealed two potential silent mutations and one differentially edited site potentially leading in about 30% of the mRNAs to a serine to glycine mutation resulting in an expression of an SRP9 protein isoform (Supplementary Fig. 9).”

Lines 368-373:

“Another analysis of a LoDEI window in the MycN-amp dataset, located in the open reading frame of the CHAF1A (Chromatin Assembly Factor 1 Subunit A) gene shows a differential editing site that is also a 3' splice acceptor site. The editing of this site could affect the splicing of the CHAF1A mRNA and further suggest that the increase in A-to-I editing associated with MycN-amplification could have an impact on CHAF1A expression (Supplementary Fig. 10).”

We added two new supplemental Figures:

Supplementary Fig. 9

Supplementary Fig. 10

Additional comments

1. The LoDEI method uses on a sliding window approach for q-value calculation. This approach partly relies on the assumption that A-to-I RNA editing regulation are spatially correlated (i.e., editing regulation of sites spatially close together in terms of genomic coordinates are correlated). If this assumption does not hold true, there might be a higher false discovery rate than presented. The authors should provide evidence of this or comment in more detail.

1.4 We thank the reviewer for this comment and apologize that we couldn't motivate the general idea clearly enough. With respect to the comment of the “spatial correlation” we like to highlight that the available A-to-I literature suggests that A-to-I editing can be both, very specific and “almost random and lead to nonselective conversion of many

adenosines” [3]. According to some publications, almost all adenosines in Alu repeats can be edited [4]. “When nucleotide positions of editing events are examined carefully among sequence reads and different samples, editing seems to fall into clusters rather than specific bases” [8]. These observations were one motivation for the development of the Alu editing index (AEI) and RNAEditor [5, 8]. LoDEI follows these ideas and also uses a non-site specific approach. As indicated in the manuscript, the major limitation of the AEI is that only a single value per sample is reported and no local information of A-to-I editing is kept. The RNAEditor follows a different concept and does not provide an analysis of differential editing. In contrast, LoDEI’s windows can be used to describe the global editing of a sample and also provide a high resolution of the genomic locations with differentially edited sites. Without LoDEI, researchers had to choose between site-specific programs like REDIT and JACUSA2 which cannot detect unspecific editing due to limited information available at single positions or the AEI which does not provide any local information. When analyzing differentially edited LoDEI windows, we find windows with only one differentially edited site and windows with many editing sites that are not necessarily at identical positions between different samples (see also response 2.3 and new Supplementary Fig. 4). To further address the concern of the reviewer we added a section to the supplementary information to better explain the implications of a window-based differential A-to-I editing detection. (Supplementary information section 3, Supplementary Fig. 17).

Changes in the revised manuscript:
See responses 2.3 and 4.3

We added the following information to the Supplementary information section 3:

Supplementary Information lines 78-101:

“Three artificial scenarios are shown in Supplementary Fig. 17 a), b), and c), to help indicating the implications of window-based approaches in comparison to a site-specific detection approaches. In all scenarios, the samples s_1, s_2, s_3 belong to set S and the samples s_4, s_5, s_6 belong to set S' . Each scenario shows three adenosines within a single window. The shown adenosines are not required to be consecutive, but need to be anywhere within a window.

Scenario a) shows an example for a site-specific A-to-I editing event. Tools like REDIT and JACUSA2 were developed to detect this kind of signal. Since LoDEI first sums up the individual signals per sample in a window and then averages across these sums, individual editing events are also detected by the window-based approach used by LoDEI as shown by the intersection analysis in the results part of the main manuscript and Supplementary Fig. 5.

In scenario b), again one adenosine is edited per sample in set S like in scenario a), but here the editing takes place at different positions instead of the same position like in scenario a). Site-specific tools like REDIT and JACUSA2 do not detect this scenario, since their statistical models require sufficient support of editing at the same position. In contrast, window-based approaches do not require a position specific editing and

call a window being differential as long as there is a difference between the windows independent of the positions in the samples.

Since the differential editing is not position-specific in a window-based approach, no differential editing would be detected in scenario c) in a window-based approach. The overall editing per sample is identical for all samples. A position-specific approach would detect all 3 positions of being differentially edited, whereas position 1 and 3 would be stronger edited in S' and position 2 would be stronger edited in S ."

We added a new supplemental Figure and legend:
Supplementary Fig. 17

2. In relation to the sliding window approach, the software provides options for specifying window size and step size for wEI and q-value calculation (lodei find). It would be useful if the authors can describe how the window and step size might impact the validity and robustness of the results, and provide recommendations for the use of non-default window and step sizes for different datasets. In addition, the authors should perform additional analyses using different combinations of window and step sizes (e.g., -w 100 -s 50, -w 200 -s 100, -w 100 -s 25) to show that the presented results are not variable when using different parameters.

1.5 We thank the reviewer for this comment. This comment is also related to a comment made by reviewer #2 (see also response 2.2). To show the impact of the window size, we added the new Supplementary Fig. 7 showing the number of detected windows as a function of the q value for 3 different window sizes. Further we tested the degree of overlap between results obtained from different window sizes. In particular we tested to what percentage the results obtained by smaller window sizes are overlapping with results obtained from larger window sizes (new Supplementary Fig. 11). In summary, more than 80% of the results obtained by smaller windows are also detected by higher values for the window size.

Changes in the revised manuscript:

We added the following sentence to the discussion of the manuscript (line 639):

"The choice of the window size affects the resolution and number of the detected windows (Supplementary Fig. 7, Supplementary Fig. 11)."

We added two new supplemental Figures and legends:

Supplementary Fig. 7
Supplementary Fig. 11

3. The authors proposed that LoDEI can detect differential editing between two sets of samples. The use of multiple biological replicates can tremendously improve the reliability and reproducibility of analysis results as it allows the estimation of signal-to-noise ratio and variability within biological groups. In the method description of LoDEI, A-to-I editing signals of all samples of a set for any given window are simply estimated

using “the mean of the editing signals of all samples of a set” (line 235). In the following step, the change in editing between two sets are calculated by taking the difference between the mean of editing signals. This approach does not seem to consider reproducibility of editing changes across replicates and could be biased by outliers. The representation of a q -value for each wEI in this context is also somewhat misleading as it might be interpreted as a measure of statistical significance of the difference between two sets of samples. The authors should implement some improvements in their software to address this, such as including a statistical test (e.g., Wilcoxon rank test) between two sets of editing signals. Otherwise, the authors should provide some comments and clarification on this issue.

1.6 We thank the reviewer for this comment and apologize that the manuscript couldn't describe the empirical q value calculation clearly enough, which caused this unfortunate misunderstanding.

“Similarly to the p value, the q value gives each feature its own individual measure of significance” [1]. A p value is a measure in terms of the false positive rate, whereas the q value is a measure in terms of the false discovery rate. For example, a false positive rate of 5% means that 5% of truly non-differentially edited A-to-I sites are called significant. Since all covered A positions need to undergo statistical testing, calling 5% of these mostly truly non-differentially edited A-to-I sites as significant yields a large number of false positives, a common problem in genome-wide testing. In contrast, a false discovery of 5% means that 5% of the detected (significant) A-to-I sites are non-A-to-I sites on average. In genome-wide testing, the quantity of interest is the q value, not the p value, since q values allow identifying “as many features in the genome as possible, while incurring a relatively low proportion of false positives”[1]. In most situations, q values cannot be estimated directly. However, in the case of A-to-I editing data, we are in the fortunate situation to be able to estimate q values directly by exploiting non-A/G mismatches for an estimation of the number of false positives, as we have done in LoDEI. The idea of exploiting non-A/G mismatches to estimate FDRs isn't entirely new and has also been used in the field of A-to-I editing [2]. Still, our described q value estimation approach has never been used for detecting differential A-to-I editing and/or in combination with a window-based approach and has never been integrated in a software for detecting differential A-to-I editing, as presented in this study. We updated the manuscript and hope to avoid misunderstandings regarding the proposed q value calculation.

Changes in the revised manuscript:

The paragraph (lines 244-248) in the first submission

“Next, we propose an empirical q value estimation for $\delta A \rightarrow G$ values based on non-A \rightarrow G differences to detect true editing events and differentiate those from false positive events. The q value provides a meaningful measure of significance for each window in such a way that a filtering of windows with a q value $\leq 10\%$ yields an overall FDR of 10% among the filtered windows.”

is changed to (lines 253-260):

“Next, we propose an empirical q value estimation for $\delta^{A \rightarrow G}$ values based on non- $A \rightarrow G$ differences to detect true editing events and differentiate those from false positive events. A q value gives each δ value its own individual measure of significance, similar to a p value. A p value is a measure in terms of the false positive rate, whereas the q value is a measure in terms of the false discovery rate and the typical quantity of interest in genome wide-testing [34]. The q value provides a measure of significance for each window in such a way that a filtering of windows with a q value $\leq 10\%$ yields an overall FDR of 10% among the filtered windows.”

4. The use of a global q -value estimation is likely heavily influenced by the overall signal-to-noise ratio or sequencing quality of the RNA-seq dataset. This might decrease the method’s sensitivity and accuracy of detecting editing changes with small effect sizes even if they are consistent. The authors should give evidence that these are true negatives or provide some comments on this.

1.7 We thank the reviewer for the comment and are pleased to see that the reviewer pointed to an advantage of the empirical q value calculation as provided by LoDEI and apologize for not stating that clearer.

Indeed, in data sets with higher noise the empirical q value estimation would correctly reflect this higher noise level, which is a feature of the program, not a drawback. If the noise level increases, smaller but true effects cannot be detected without the cost of increasing more false positive detections as well. In this scenario, smaller effect sizes would get a higher q value. Consequently, the decision which q value threshold to use is up to the individual researcher. We believe it would be hard to argue in favor of a perfect sensitivity at the cost of detecting more false positives than true positives.

Supplementary Table 1 showed that LoDEI identified different thresholds with an FDR of 10% for each dataset. Different thresholds indicate different noise levels in the analyzed datasets. The table below is an excerpt from supplementary table 1:

Dataset	Threshold for $\delta^{A \rightarrow G} < 0$	Threshold for $\delta^{A \rightarrow G} > 0$
ADAR KD	-11.1	Inf
Ro60 KO	-Inf	16.6
MycN-amp	-Inf	30.4
sncRNA7SL OE	-23.2	Inf

We believe that the changes to the manuscript with respect to the responses 1.8, 2.10 and 2.11 help to avoid misunderstandings with the proposed q value estimation.

5. In the Results section 2.1 line 269-273, the statement that “we assume increasing q values for strong values of δ as counterintuitive and avoid varying q value estimates by

not allowing those estimates to increase again” is very vague and unsupported. The authors should provide some statistical/mathematical evidence or clarification for this step.

1.8 We thank the reviewer for the comment and apologize for not explaining our rationale more clearly. Our assumption is based on the negative correlation between the number of available windows and $\delta^{A \rightarrow G}$ values. The stronger the $\delta^{A \rightarrow G}$ values, the smaller the number of available windows to estimate an empirical q value. The decreasing number of available windows for the q value estimation can lead to a high variance of the q value estimates for extreme values of $\delta^{A \rightarrow G}$. To better explain this phenomenon, we added the new Supplementary Fig. 8 and added an additional explanation to section 2.1. Typically, the q value decreases the stronger the values for $\delta^{A \rightarrow G}$ get. For extreme values of $\delta^{A \rightarrow G}$, where the variability of the q value estimate increases, situations can arise where higher signals get higher q values again, while lower values of $\delta^{A \rightarrow G}$ already had lower q values. These situations are caused by a limited number of available windows and not by underlying biology. Thus, we don't allow q values to increase again.

Changes in the revised manuscript:

The paragraph (lines 269 - 273) in the first submission

“The stronger the calculated δ values by LoDEI, the smaller the number of those observed differences that can lead to strong variability of the estimated q values if the number of observed δ values is small. We assume increasing q values for strong values of δ as counterintuitive and avoid varying q value estimates by not allowing those estimates to increase again.”

was changed to (lines 281-290)

“The stronger the calculated δ values by LoDEI, the smaller the number of those observed differences. This negative correlation can lead to strong variability of the estimated q values if the number of observed δ values is small. Typically, the q value decreases the stronger the values for $\delta^{A \rightarrow G}$ get. For extreme values of $\delta^{A \rightarrow G}$, where the variability of the q value estimate increases, situations can arise where higher signals get higher q values again, while lower values of $\delta^{A \rightarrow G}$ already had lower q values (Supplementary Fig. 8). These situations are caused by a limited number of available windows and not by underlying biology. Consequently, assume increasing q values for strong values of δ as counterintuitive and avoid varying q value estimates by not allowing those estimates to increase again.”

We added one new supplementary Figure and legend:
Supplementary Fig. 8

6. The authors should provide more biological interpretation of the new results in this paper. With the identified novel differential editing sites and novel regulators of A-to-I editing, the authors should investigate and elaborate on their roles and functions in biological functions and how their dysregulation might be implicated in diseases such as cancer or autoimmune disease. The authors should also perform additional differential RNA editing analyses in disease model or patient sample RNA-seq data to validate their novel results to more clearly demonstrate the biological significance of their new findings.

1.9 As appreciated by the reviewer we present some novel data on regulators of A-to-I editing. Of course, the biological interpretation of those findings and validation of biological effects are the next exciting step after the identification of changes in A-to-I editing. As described in the introduction, the cellular regulation and experimental modulation of A-to-I editing is of growing interest in basic and applied science. To point out some potential biological implications of this study, we presented the new supplemental Figures 9 and 10 as detailed in the response 1.3 (above).

The analysis of patient data sets is of course of significant interest also for us. However, the analysis of complex patient data sets and more importantly, the analysis of the biological implications and experimental validation are well beyond the scope of this manuscript.

Reviewer #1 (Remarks on code availability):

The README on the GitHub repository is written clearly with instructions to run the software. I tested the software installation using conda and pip and was able to successfully run the test data provided. However, the authors should include more details of the results in the Output section, such as what columns are expected in the output files and what they mean, how merged scores were calculated etc.

1.10 We thank the reviewer for testing the software and are thankful for the suggested improvements for the output section of the software documentation. We added additional information about the outfiles provided by LoDEI to the corresponding GitHub page for an easier usage of the program.

Reviewer #2 (Remarks to the Author):

The authors present a new method for testing for differential A-to-I editing from RNA sequencing data. The main novelty on the approach lies in the use of a sliding window, which can potentially increase power by capturing editing sites in groups instead of considering them singularly (also biologically relevant given ADAR's indiscriminate editing behaviour in double stranded regions). They further incorporate non-parametric testing of differential editing by considering a null-distribution of change observed

across non A-to-G edit types, test on 4 datasets and find that their method generally detects more differentially edited sites compared to JACUSA2 and REDIT.

Overall, developing novel approaches (sliding window/ non-parametrics) in this context is useful given there currently lacks methods for differential A-to-I editing. However, to be adopted by the community LoDEI requires a lot more testing with regards to the usefulness/robustness of the approach, see comments below.

> By taking the difference of two averages instead of the average of the differences in the window you are removing the paired information of the individual sites - could this result in a loss of statistical power? Some discussion of this and more details regarding the reasoning behind the methodological decision making would be useful. Also, by summing sites, the inherent variability in editing across the replicate samples is lost - some discussion of this aspect is required

2.1 We thank the reviewer for this comment and agree that utilizing a window-based instead of a site-specific approach has implications about which signals can be detected by the different approaches and apologize that we missed to explain the differences between the different concepts clearly enough.

We added a more detailed explanation about the implications of site-specific and our window-based approaches to better explain the differences between the two concepts to the supplementary information (Supplementary information section 3, Supplementary Fig. 17).

See also answer 1.4 for an explanation about the general motivation of the proposed window-based approach.

Changes in the revised manuscript:

We added a new section to the supplementary information (lines 76-101):

We added one new supplementary Figure:
Supplementary Fig. 17

> The 'sliding window approach' is not well explained in the methods/text - how exactly are the windows calculated, are they non-overlapping (like depicted in Figure 1B)? What is the size of the window (the chosen 'w') for the various studies and how did the authors come forward to the most appropriate 'w' for each? Reading the instructions on the Github page actually did help figure out the defaults, but more needs to be said in the text.

2.2 We thank the reviewer for pointing out the shortcomings in explaining the proposed sliding window approach. We changed the manuscript and the implementation to improve the manuscript and the program's usage.

In the updated software version, we removed the step-size option (-s). Instead, the program determines the step size of a window from the given window size so that exclusively non-overlapping windows, as depicted in Fig 1b and as used in the analysis of the manuscript, are used in the differential editing index calculation.

For a discussion of the window size we redirect the reader to 1.5.

Changes in the revised manuscript:

We added the following sentences to the main manuscript:

Section 2.1 line 237:

“Note, LoDEI uses non-overlapping windows with a default size of 51 nucleotides.”

Section 4.3 line 800:

“If not stated otherwise, programs were run with default parameters.”

> What is the overall distribution of the numbers of sites in the windows transcriptome wide (are there many containing singletons or does it generally capture clusters)? Can different 'information' be obtained by using different window sizes in the same comparison?

2.3 We thank the reviewer for this comment and apologize that we missed to show the number of A-to-I sites in detected windows and didn't describe the information that is captured by a window-based approach clearly enough in the first submission. We added an analysis to show the number of sites per window in the new Supplementary Fig. 4 and the new Supplementary Fig. 14, and added an explanation about the implications of a window-based approach (Supplementary information section 3, Supplementary Fig. 17). A window-based approach as used in LoDEI can detect editing not occurring at the same genomic position which is unfeasible by design in single site approaches. Thus, the size of the window should be chosen with this in mind. It's a decision between a higher resolution and the availability to detect non-site-specific editing. If the size is too small, non-site-specific editing won't be detected.

In this context please see also our response to point 1.5.

Changes in the revised manuscript:

We added in line 502 to section 2.4 in the main manuscript:

“LoDEI's detected windows can contain both, single and clusters of A-to-I editing sites (Supplementary Fig. 4, Supplementary Fig. 14).”

We added three new supplementary Figures and legends:

Supplementary Fig. 4
Supplementary Fig. 14
Supplementary Fig. 17

> It is claimed that LoDEI is applicable for comparisons between single sample, but I do not see any convincing demonstrating/testing of this idea by comparing the results for 1-2-3 samples from the test datasets.

2.4 We thank the reviewer for this comment and apologize that the claims in the manuscript haven't been supported by further analysis in the first submission. We added an analysis of the comparison of the results obtained from the comparison of single samples to the supplementary information and refer to that in main manuscript (new Section 2 in the supplementary information, and new Supplementary Fig. 15 and 16). In brief, we used the Jaccard index to calculate the similarity between detected differential A-to-I regions obtained from the comparison of single samples and refer the reader to section 2 of the supplementary information for a detailed description.

Changes in the revised manuscript:

Line 516:

“In experiments with strong A-to-I editing changes, LoDEI can detect differential A-to-I editing between single samples (Supplementary Fig. 15, 16).”

We added in lines 53 – 75 to the supplementary information section 2:

“To test if LoDEI might be able to detect differential A-to-I editing between single samples, we first generated results using LoDEI for each pairwise comparison between individual samples of sets S and S' . When naming individual samples only by their numeric index starting at 0, we can use the numeric indices of both samples to indicate which samples were used to generate the results. For instance, '01' is the name of the LoDEI result of the comparison of sample s_0 from set S with sample s_1 from set S' (Supplementary Fig. 15). We use this naming scheme in Supplementary Fig. 16.

To compare the detected differential A-to-I editing obtained from the pairwise comparisons, we used the Jaccard index. The Jaccard index $J(A, B)$ measures the similarity of two sets A and B by dividing the intersection of A and B by the union of A and B :

$$J(A, B) = \frac{A \cap B}{A \cup B}$$

Since LoDEI reports windows and not single positions, we first generated a list of all genomic positions covered by the reported windows which yields the sets A and B . The generation of these lists is necessary to calculate a precise Jaccard index. Without transforming the windows into single positions, it would be questionable what an overlap between two windows is. For instance, an overlap of two windows by only a single position could be considered as an overlap of the results which could yield a larger overlap. To avoid such a potential bias and report a position-specific comparison of overlaps of windows, we first generate a list of the covered genomic positions and calculate the Jaccard index based on those genomic positions (Supplementary Fig. 16).

A detection of differential A-to-I editing detection based on the comparison of single samples could only be achieved in the ADAR KD and Ro60 datasets where a strong difference in A-to-I editing is known.”

We added two new supplementary Figures:

Supplementary Fig. 15

Supplementary Fig. 16

> Figure 2. The y scale can vary a lot, down to -300 in some cases. I guess it depends on the expression at the individual site, but wouldn't some kind of normalised score make sense?

2.5 We thank the reviewer for this comment. The strong differences in the y scale are caused by the ADAR KD that causes extrem strong changes in the A-to-I editing in many cases. This dataset is an ‘outlier’ compared to the other 3 datasets. Since the editing at a single position is defined as the observed a A/G mismatches divided by the coverage at that position (Eq. 1) the values are already in a normalized form.

> Use of the threshold $q < 0.1$ - would be helpful with more consideration/discussion about how to tune the threshold for capturing the most biological relevant changes.

2.6 We thank the reviewer for this comment. Since the q value is a measure of the expected fraction of false positives, a q value threshold is highly subjective. It remains a question for an individual researcher or experimental setting to decide which fraction of false positives is acceptable. A threshold for q might vary between different research questions. Users can filter the results according to their needs. In general, the lower the q value, the higher the observed differential editing signal.

> Figure 3: For the ADAR KO data, one would specifically expect many of the significantly reduced sites upon KO to be supported by REDportal - what is the precision/recall of the LoDEI method with respect to REDportal (which I realise is not perfect but still highly relevant), and compared to the other other methods? Also, some discussion of how JACUSA/REDIT actually work and differ in terms of their methods, and how they were applied to be optimally comparable with LoDEI is important.

2.7 We thank the reviewer for the suggestion to compare identified differential A-to-I sites with information stored in the REDportal database. We agree that such a comparison helps to interpret the derived results, so we added a comparison in terms of the percent overlap of detected A-to-I editing with the REDportal as a function of the q value to the main manuscript (new Fig. 5).

Using the percent overlap with the REDportal describes the support of the detected sites by the database and avoids treating the REDportal as ground truth as it would be needed to be when explicitly using the terms precision and recall. The REDportal is a great resource but contains A-to-I events from many different tissues, while the sites shown in the manuscript are derived from single cell lines, which makes a definition of a recall problematic. Detected differential A-to-I sites that are not supported by the REDportal are also not necessarily false positives. As an example, the A-to-I editing site shown in Supplementary Fig. 10 was not found REDportal. Thus, we decided to use the phrase “percent overlap” instead of calling it precision and show the overlap as a function of the q value (Fig. 5).

We added an explanation of how JACUSA and REDIT were applied to be comparable to LoDEI to section 2.4.

Changes in the revised manuscript:

We added a new section 2.6 “Differential A-to-I events detected by LoDEI overlap with known editing sites in the REDportal” to the main manuscript (lines 567-592).

“We tested whether A-to-I events detected by LoDEI are listed in the REDportal database to provide further evidence for real differential editing events. As of this writing, the REDportal consists of around 16 million A-to-I sites based on 9642 human RNAseq samples from 31 different tissues of the GTEx project [23].

In general, for small q value thresholds, most detected A-to-I events by LoDEI overlap with editing sites listed in the REDportal, and the percent of overlap decreases with increasing q value thresholds (Fig.5). At a q value threshold of 0.05, 75.9%, 78.1%, 65.3%, and 93.8% of A-to-I events detected by LoDEI in the ADAR1 KD, Ro60 KO, MycN-amp, and snRNA7SL OE datasets respectively, are overlapping with the REDportal database. Of note, since REDIT and JACUSA2 did not detect differential A-to-I editing in all but the ADAR KD dataset, only minor or no overlap of the REDIT and JACUSA2 datasets with the REDportal database was found, except for the ADAR KD dataset.

Next, we performed the REDportal overlap analysis for the different genomic locations (Fig. 6). Here, a notable difference in the percent overlap of A-to-I events in the 5'UTR and exon regions between the sncRNA7SL and the other datasets can be observed. In the sncRNA7SL dataset, almost all detected differential sites are listed in the REDportal compared to the other three datasets.”

We added in lines 713-716 in the discussion of the main manuscript:

“The large overlap of detected differential A-to-I events with editing sites listed in the REDportal further supports our results. Of note, detected differential A-to-I editing not overlapping with REDportal data does not indicate false positive detection as exemplified (Supplementary Fig. 10).”

We added lines 494-497 to section 2.4:

“To obtain q values from REDIT and JACUSA2 for a direct comparison with results from LoDEI, we used a similar procedure as utilized in LoDEI. Therefore, REDIT and JACUSA2 were applied on G/A mismatches to approximate the number of false positives to finally calculate q values for detected differential A-to-I editing.”

We added two new Figures and legends to the main manuscript::

Fig. 5

Fig. 6

>Figure 4 ditto - how many of the sites found in b-d are editing sites supported by REDportal? And in 4a) how many sites/regions are shared by the methods, and are the shared ones driven by larger changes in editing proportion?

2.8 We thank the reviewer for the request to provide the support of detected A-to-I editing by the REDportal. We added the missing analysis and provide the percent overlap of detected A-to-I editing with the REDportal in Fig. 6 of the updated manuscript (see 2.7). Detected A-to-I editing is supported to a large extent by editing sites listed in the REDportal.

Changes in the revised manuscript:
see 2.7

We apologize that our description in the results part that describes the sharing of sites/regions detected by different methods hasn't been detailed enough in the first version of the manuscript. To fully describe the overlaps between the different methods we added an upset like plot that shows the overlaps between sites and region between the different methods (Supplementary Fig. 5). Note that due to the comparison of site-based and LoDEI's window-based approaches the intersection between programs

needs to be given twice: from the perspective of the windows and from the perspective of single sites.

We added a new Supplementary Fig. 6 that shows the distribution of $\delta^{A \rightarrow G}$ of shared and unique windows.

Changes in the revised manuscript:

We added the references to Supplementary Fig. 5 and Fig. 6 in section 2.4 of the main manuscript.

We added new Figures and legends to the supplemental section:

Supplementary Fig. 5

Supplementary Fig. 6

> Much of the data used in the paper likely has very large shifts in ADAR explaining many of the changes/global shifts in editing. Showing extra supporting figures to show the direction of ADAR change would be useful in the context of understanding the results.

2.9 We thank the reviewer for this comment and added a differential gene expression analysis using DESeq2 for all data sets to report the change of the gene expression of ADAR proteins for all analyzed data sets (Supplementary Table 2).

Note, that “ADAR expression levels are not necessarily correlated well with A-to-I RNA-editing levels of target RNAs within a given tissue or developmental stage”[6]. This is likely due to the regulation of A-to-I editing by other factors as found herein and described in the manuscript. Interestingly, we found that in the Ro60, MycN and sncRNA7SL datasets, ADAR2 expression levels were significantly changed. The reason for this very interesting biological observation is currently not understood.

Changes in the revised manuscript:

We added line 302 in section 2.2:

“All datasets show a significant change of the gene expression in at least one of the genes of the ADAR family (new Supplementary Table 2).”

We added one new Supplementary Table and legend:

Supplementary Table 2

> Since C>U can be valid editing (mediated by APOBEC) it seems strange to include it in the list of possible background noise (though I acknowledge that the median strategy might filter this, but more discussion would be useful).

2.10 We thank the reviewer for this comment and agree that C>U changes can be valid editing. We apologize for not explaining why we included known non-A-G editing signals in the empirical q value calculation in the first submission. We added the missing information to the current version of the manuscript and hope that the motivation is now clearer for the reader. As the reviewer correctly states, the motivation to take the median is to avoid the effects of potentially strong outliers. The motivation to keep q value estimates for all non-A-G differences is to construct a rather conservative estimate. With keeping all non-A-G based q value estimates for the final estimation, those estimates will tend to report rather higher than lower q values and thus yielding a more conservative estimate.

Changes in the revised manuscript:

We added in line 717-721 in the discussion of the main manuscript:

“For the empirical q value calculation, we decided to use all non-A \rightarrow G -mismatches, even knowing that some non-A \rightarrow G mismatches like C \rightarrow T mismatches can be valid editing caused by other proteins than ADAR. With keeping all non-A \rightarrow G based q value estimates, the final q values for A \rightarrow G mismatches will tend to be rather higher than lower q values and thus being conservative estimates.”

> Treatment of SNPS appears somewhat downplayed in the focus of the article but one can imagine the existence of SNPs compared to the reference would really inflate the background null, including heterozygous SNPs which are even less obvious to see. More description of the methods and discussion about the steps taken to deal with SNPs should be included.

2.11 We thank the reviewer for the comment and are pleased to see that the reviewer pointed out the advantage of the empirical q value calculation as provided by LoDEI and changed the manuscript accordingly to make this point more apparent to the reader. In experiments where cell lines are compared against each other (e.g., the ADAR KD, Ro60 KO, and sncRNA7SL datasets), SNPs do not have an effect on LoDEI's $\delta_{S,S'}^{X \rightarrow Y}$ values. SNPs are part of every sample in both sets and do not impact on the difference between the sets. In scenarios where the compared sets consist of different types of cells, SNPs indeed have an effect. However, non-A-G data used for the empirical q value calculation is affected in the same way and correctly informs the user about the expected number of false positives in the analysis. In cases where SNPs start affecting the data heavily (i.e., haven't been removed), the empirical q values correctly reflect this situation in the provided results and q values will increase.

Changes in the revised manuscript:

We changed the SNP paragraph in the discussion in line 684-698 in the main manuscript:

“ Any software that utilizes observed mismatches from NGS data can be affected by SNPs. Note, that in experiments where cell lines are compared against each other (e.g., the ADAR KD, Ro60 KO, and sncRNA7SL datasets), SNPs do not affect LoDEI’s $\delta A \rightarrow G$ values, since SNPs are part of every sample in both sets and do not impact the difference between the sets (Eq. 3, Supplementary Fig. 17). However, in scenarios where the compared sets consist of different types of cells, SNPs can have an effect that is correctly reflected by the empirical q value estimation. The non-A \rightarrow G mismatches used for the empirical q value estimation are affected in the same way and thus also contain a similar amount of SNPs. As a consequence, empirical q values will start to increase and correctly inform the user about the expected number of false positives in the analysis. In cases where SNPs start affecting the data heavily (i.e., haven’t been removed), the empirical q values correctly reflect this situation in the provided results and indicate that SNPs should be treated explicitly upfront in the analysis pipeline.”

Reviewer #2 (Remarks on code availability):

Note: I have read the GitHub front page which appears well explained but do not have available time to run the code myself.

We thank the reviewer for the acknowledgment of the carefully written documentation.

Reviewer #3 (Remarks to the Author):

We thank the co-reviewer for contributing to the comments brought up by one of the other reviewers and wish all the best for the next steps in the research career.

Reviewer #4 (Remarks to the Author):

In this article, Torkler et al. presented a novel method to more sensitively detect differential A-to-I editing in RNA-seq datasets based on a sliding-window approach coupled with an empirical q-value calculation. The authors called this method the “local differential editing index” (LoDEI). The LoDEI approach was compared with results from the site-specific differential A-to-I detection tools, such as REDIT and JACUSA2, as well as with results of the global A-to-I detection as provided by the Alu Editing Index (AEI).

Although the results presented by the authors are interesting, I have several comments that should be addressed before being considered for publication. My comments are detailed below.

Major comments:

- One of the major points of discussion is the authors' choice of the window length, denoted as 'w'. It needs to be made clear how this value is determined. Is it a user-defined parameter? The choice of the window length is a crucial aspect of the LoDEI calculation, and it would be beneficial for the authors to explain their rationale. Additionally, it would be insightful to see how varying the window length impacts the number of significant differences.

4.1 We realized that the “sliding windows approach” led to comments by all reviewers and regret, for not explaining this approach more clearly initially. Yes, the size of the window is a user-defined parameter. The larger the windows are, the less position specific information is kept. Thus, at the end the user needs to decide which resolution is needed. We added this information to the discussion of the manuscript please see also answers 1.5, 2.2, and 2.3. New supplementary Figures 4, 7, 11, 14, and 17 address the „windows“ comments:

Changes in the revised manuscript and please see response 1.5:

We added the following sentence to the discussion of the manuscript (lines 639-640):

“The choice of the window size affects the resolution and number of the detected windows (Supplementary Fig. 7, Supplementary Fig. 11).”

We added one new supplemental Figure and legend:

Supplementary Fig. 7

Changes in the revised manuscript and please see response 2.2:

We added the following sentences to the main manuscript:

Section 2.1 line 237:

“Note, LoDEI uses non-overlapping windows with a default size of 51 nucleotides.”

Changes in the revised manuscript and please see response 2.3:

We added in line 500-501 to section 2.4 in the main manuscript:

“ LoDEI's detected windows can contain both, single and clusters of A-to-I editing sites (Supplementary Fig. 4, Supplementary Fig. 14).”

See Supplementary Information section 3 for implications of a window-based differential editing calculation

- Following my comments above, regarding the length of the window, is there any difference between samples from cell lines and tissue samples? Also, as the RNA editing phenomenon depends on RNA structures, in which repetitive elements play an essential role, how does the length of the window impact different species, such as mice and drosophila, compared to humans? Finally, what is the size of the window chosen for the analyses performed in the study, and what is the justification for such a choice?

4.2 We thank the reviewer for these questions. The implications of a window-based differential A-to-I editing detection is described in the responses to the window size topic (see response 1.4, 1.5, 2.1, 2.2). We do not see a reason why the window-based approach could result in a different outcome if the RNA is prepared from tissue or cell lines. However, we can appreciate that it is very likely that the A-to-I editing of a specific mRNA can change when comparing data from cell lines or tissue, but this is likely based on the complex and different regulation of A-to-I editing in cell lines and tissue and not on the LoDEI based calculations. As shown in Supplemental Figures 1 and 2, differential A-to-I editing in Alu elements is shown, whereas in Supplemental Figures 9 and 10 differential A-to-I editing is found in exons without Alu elements, demonstrating that LoDEI detects differential editing in different kind of sequence context. To test LoDEI's general applicability further - as suggested by reviewer 4 - we tested the program on a *C. elegans* dataset (see response 4.4).

- At the beginning of Section 2.4, lines 451 and 452, the authors stated that besides LoDEI, no window-based approach exists, including a statistical framework for differential A-to-I editing detection. However, I would recommend considering these two previously published tools, such as RNAEditor and FLARE (PMIDs: 27694136, 37784060), which use a "window-" or "cluster-"based approach and see if there are any differences or similarities with the LoDEI tool. The authors could also assess if these tools are suitable for detecting RNA editing differences between different conditions. If not, at least they could compare them with LoDEI regarding detecting A-to-I editing sites.

4.3 We thank the reviewer for highlighting other “window- or cluster-based approaches” and apologize for not mentioning them explicitly in the first submission. In contrast to LoDEI, REDIT, and JACUSA2 RNAEditor and FLARE are a different class of A-to-I software whose purpose is for the detection of any A-to-I signal, and they are not designed for the detection of differential A-to-I editing and do not provide any statistical model for detecting differential editing. Thus, neither RNAEditor nor FLARE provides valid data for a direct comparison. LoDEI, REDIT and JACUSA2 were explicitly developed to address the question of detecting differential A-to-I editing and providing a corresponding statistical model. RNAEditor and FLARE are competitors to A-to-I detection software like REDIttools or GIREMI, but not to differential A-to-I detection software like LoDEI, REDIT, and JACUSA2.

We added the missing information to the introduction of the manuscript.

Changes in the revised manuscript:

In the introduction we changed

“To address the drawbacks of the detection of single sites probably caused by the widespread binding of ADAR1 to dsRNA, the Alu editing index (AEI) does not focus on single A-to-I sites but computes the ratio of the number of A → G mismatches and total coverage in predefined Alu regions to report the A-to-I editing [25]. The AEI is a single global value for a complete sample and does not report any positional information of local editing changes.”

into:

“To address the drawbacks of the detection of single sites probably caused by the widespread binding of ADAR1 to dsRNA alternative approaches like RNAEditor, FLARE and the Alu editing index (AEI) have been proposed that share the common idea to detect A-to-I editing by analyzing a larger genomic region rather than analyzing single nucleotides [25–27].

~~The Alu editing index (AEI) does not focus on single A-to-I sites but computes the ratio of the number of A/G mismatches and total coverage in predefined Alu regions to report the A-to-I editing. The AEI is a single global value for a complete sample and does not report any positional information of local editing changes.”~~

The deleted information has been edited and moved to the results section 2.3 where the AEI results are compared with LoDEI.

In 2.3 we changed

“The AEI reports a single value for each sample that summarizes the global amount of A-to-I editing.”

To (lines 397-399)

“The AEI reports a single value for each sample that summarizes the global amount of A-to-I editing of a sample and thus does not keep any positional information of the A-to-I editing events.”

- The authors choose four different datasets to compare LoDEI with other methods, such as REDIT and JACUSA2. These are suitable datasets to make these comparisons; however, they are limited as they are derived from human cell lines. As REDIT and JACUSA2 tools and others were employed in several other sample data, I would suggest having a more comprehensive comparison with human tissue samples, for example, data from TCGA, TARGET, or GTEx (even choosing a tissue type as a case study), or different species (e.g., mouse, drosophila).

4.4 We thank the reviewer for the comment and the confirmation that the chosen datasets are suitable for the comparisons of the different tools. To further support the claims of the manuscript we added an analysis of a *C. elegans* ADAR mutant dataset to the manuscript (Supplementary information section 1, Supplementary Fig. 12, 13, 14).

Changes in the revised manuscript:

We added the following paragraph to section 2.2 (lines 325-329) of the main manuscript:

“To further support LoDEI ’s general applicability we analyzed differential A-to-I editing in non-human datasets. We analyzed RNA-seq data from ADAR mutant and wildtype *C. elegans* worms and observed the same consistent contrast between A → G and non-A → G differences as in the human datasets (Supplementary Fig. 12). Strong editing differences are almost exclusively observed for A → G differences.”

We added section 1 to the supplementary information containing the *C. elegans* analysis (lines 42 – 53).

We added three new supplemental Figures and legends:

Supplementary Figure 12

Supplementary Figure 13

Supplementary Figure 14

- The authors should put more information about the mapping (including the parameters used for the STAR tool) and RNA modification annotation steps. For example, they specified that the filtering of SNPs was only employed for the MycN-amp dataset. Such a filter is an essential step in the RNA editing characterization, and I recommend including it as a mandatory step in the workflow. Moreover, several of the RNA editing detection tools, to reduce the number of false positives in the modification events characterization, employ Bayes models or Binomial tests based on the quality base of nucleotide with editing event (information present in the FASTQ files) and filtering those RNA editing events with low score or not significant. I suggest the authors include this aspect in the modification event characterization workflow to reduce the number of false positives. Several other filters used by other tools can help reduce the false positives in RNA editing detection, and it would be helpful to at least discuss some of them in the text and adopt them in the workflow.

4.5.1 We thank the reviewer for the comments and included additional information in the online methods section of the manuscript. In addition, we like to point to the provided container images that contain the complete pipeline used to derive the results and as such also contain all used parameters.

Changes in the revised manuscript:

We added in line 799 in the online method section:

“If not stated otherwise, programs were run with default parameters.”

“...using STAR with default parameters” (line 806)

4.5.2 Reviewer #2 also commented on A-to-I detection with respect to SNPs. We regret, for not explaining this problem more clearly initially. Please see our response in 2.11 for details about SNPs:

4.5.3 We thank the reviewer for the remarks regarding mapping quality and added the information to the manuscript that proper quality filtering of the sequencing reads, as was done in the shown analysis, should be performed prior to A-to-I analysis to reduce the number of false positives. Note, since the q values in LoDEI are calculated empirically the q values would increase accordingly with respect to the higher amount of sequencing errors. All potential error sources mentioned would also occur in the non-A-G mismatches and as such directly affect the q value estimation. Thus, the empirical q value calculation would correctly reflect these situations and inform the user accordingly.

Changes in the revised manuscript:

We added in lines 802-804 in the online method section:

“To reduce the chance of false positives, we recommend proper quality filtering of the sequencing reads prior to differential A-to-I detection.”

Minor comments:

- The authors should try to explain why they used these datasets and give an idea of the expected outcomes from the comparisons in the first paragraph (lines 279 to 283). This was done in Section 2.2.1, but it would be better to anticipate it so the reader could easily follow the flow of the results presented in Figure 2.

4.6 We thank the reviewer for this comment and agree that a reordering of the statements enhance the readability of the manuscript, and we moved the explanations of the datasets from 2.2.1 to section 2.2.

Changes in the revised manuscript (lines 300-314):

“The first two of the four datasets are known to show differential A-to-I editing and are used to show the general applicability of LoDEI’s approach. All datasets show a significant change in the gene expression in at least one of the genes of the ADAR family (Supplementary Table 2).

The first analyzed RNA-seq dataset consisting of two samples per set is known to show a strong reduction of A-to-I editing upon siRNA-induced ADAR1 knockdown (KD) in the glioblastoma cell line U87MG when compared to a control group [16]. Other RNA-binding proteins are known to regulate A-to-I editing besides ADAR, and several

publications could show an increase in A-to-I editing upon the reduction of Ro60 (TROVE2) [24, 37, 38]. Hence, we used the RNA-seq dataset consisting of two control and three knockout samples derived from the Ro60 knockout (KO) Lymphoblastoid cell line GM12878 as the second dataset [17].

After evaluating the general applicability of LoDEI on the ADAR KD and Ro60 KO datasets, we applied LoDEI to novel datasets to search for differential A-to-I editing.”

- In lines 671 and 672, the authors stated, “a position is excluded from the calculation, if a single position in any of the samples of a set shows a mismatch frequency 80%.” However, this needs a reference or at least justification.

4.7 We thank the reviewer for this comment and added the missing information that the above-mentioned filtering was used as a simple ad-hoc filtering of potential SNPs. For further SNP related information please see 2.11.

We added in lines 812-813 in the online method section:

“As an ad-hoc filter to remove potential SNPs,…”

References:

[1] Storey JD, Tibshirani R. Statistical significance for genomewide studies. *Proc Natl Acad Sci U S A*. 2003;100(16):9440-9445. doi:10.1073/pnas.1530509100

[2] Ramaswami G, Zhang R, Piskol R, et al. Identifying RNA editing sites using RNA sequencing data alone. *Nat Methods*. 2013;10(2):128-132. doi:10.1038/nmeth.2330

[3] Nishikura K. Functions and regulation of RNA editing by ADAR deaminases. *Annu Rev Biochem*. 2010;79:321-349. doi:10.1146/annurev-biochem-060208-105251

[4] Bazak L, Haviv A, Barak M, et al. A-to-I RNA editing occurs at over a hundred million genomic sites, located in a majority of human genes. *Genome Res*. 2014;24(3):365-376. doi:10.1101/gr.164749.113

[5] Roth SH, Levanon EY, Eisenberg E. Genome-wide quantification of ADAR adenosine-to-inosine RNA editing activity. *Nat Methods*. 2019;16(11):1131-1138. doi:10.1038/s41592-019-0610-9

[6] Nishikura K. A-to-I editing of coding and non-coding RNAs by ADARs. *Nat Rev Mol Cell Biol.* 2016;17(2):83-96. doi:10.1038/nrm.2015.4

[7] Bahn JH, Lee JH, Li G, Greer C, Peng G, Xiao X. Accurate identification of A-to-I RNA editing in human by transcriptome sequencing. *Genome Res.* 2012;22(1):142-150. doi:10.1101/gr.124107.111

[8] John, D., Weirick, T., Dimmeler, S., & Uchida, S. (2017). RNAEditor: easy detection of RNA editing events and the introduction of editing islands. *Briefings in bioinformatics*, 18(6), 993–1001. <https://doi.org/10.1093/bib/bbw087>

[9] Liddicoat, B. J., Piskol, R., Chalk, A. M., Ramaswami, G., Higuchi, M., Hartner, J. C., Li, J. B., Seeburg, P. H., & Walkley, C. R. (2015). RNA editing by ADAR1 prevents MDA5 sensing of endogenous dsRNA as nonself. *Science (New York, N.Y.)*, 349(6252), 1115–1120. <https://doi.org/10.1126/science.aac7049>

[10] Gatsiou, A., Tual-Chalot, S., Napoli, M., Ortega-Gomez, A., Regen, T., Badolia, R., Cesarini, V., Garcia-Gonzalez, C., Chevre, R., Ciliberti, G., Silvestre-Roig, C., Martini, M., Hoffmann, J., Hamouche, R., Visker, J. R., Diakos, N., Wietelmann, A., Silvestris, D. A., Georgiopoulos, G., Moshfegh, A., ... Stellos, K. (2023). The RNA editor ADAR2 promotes immune cell trafficking by enhancing endothelial responses to interleukin-6 during sterile inflammation. *Immunity*, 56(5), 979–997.e11. <https://doi.org/10.1016/j.immuni.2023.03.021>

REVIEWERS' COMMENTS

Reviewer #1 (Remarks to the Author):

In the revised manuscript, the authors have provided additional biological interpretations of their results and clarifications on the scientific rationale behind the proposed method. However, the manuscript still lacks convincing evidence that the higher sensitivity of LoDEI is detecting true and biological meaningful editing sites. By aggregating replicates and using a genome-wide false discovery rate calculation approach, the method is potentially detecting a high number of false positives in their newly identified editing sites. If this is the case, LoDEI does not out-perform any of the current published methods and the use of its results is potentially detrimental to a study as it might lead to misinterpretation of biological outcomes. Without any experimental validation, simulation data, or cross validation using additional RNA-seq datasets, the authors have not provided any additional evidence to validate their claim that LoDEI is more accurate than other methods, albeit being able to detect a higher number of editing sites. Overall, the revised manuscript lacks evidence that LoDEI is an improved method for differential editing site detection.

Specific points:

1. The authors should perform experimental validation of their findings using a low throughput method to demonstrate that the method can detect true editing events missed by other methods. Otherwise, the authors should provide some cross-validation of their findings using additional RNA-seq datasets. For example, the authors can show that the detection of new differential editing sites can be reproduced reliably in multiple datasets with the same expected editing outcomes.
2. It has been pointed out by several reviewers that the aggregation of replicates, by taking the mean, leads to the loss of statistical power in detecting differential editing sites as the variance is not considered. This point has not been addressed by the authors. The use of a global q-value is not sufficient when considering highly variable sites, e.g., sequencing artefacts. The authors should address and elaborate on this in detail.
3. Although the authors show a high degree of overlap between their detected A-to-I editing sites and the REDportal database, they have yet to demonstrate that the additional sensitivity provided by LoDEI has detected new true editing events and not false positives. The high overlap does not address the potential problem of a high false positive rate. In addition, this comparison does not consider differential editing events, which is presented as a main feature of LoDEI, but only the detection of basally edited sites. A visualisation of a select few editing sites in the same dataset that LoDEI was tested on is “double-dipping” and does not suffice as validation of their findings.
4. In supp fig. 7 and supp fig. 11, although the authors show that the overlap of the intersection across the tested window sizes are >80%, there appears to be large variability in LoDEI's sensitivity of detection of differential editing windows. This is potentially a major drawback of the method as its sensitivity can be heavily influenced by the choice of its run parameters. Although the authors have added a comment on how the window size affects the resolution, they have not provided any information or empirical evidence on their (or the user's) choice of window size. The authors should address this with additional data or a very detailed discussion.

Reviewer #2 (Remarks to the Author):

I believe the authors have done a thorough job in addressing my comments (and those of the other reviewers).

One small point relating to my previous comment about sample variability. The authors have clearly demonstrated the utility of their empirical-based method, but I think it is still worth a mention in the Discussion that a limitation is that it does not take into account the sample-sample editing variability within each condition.

Reviewer #3 (Remarks to the Author):

Reviewer #3 (Remarks on code availability):

I did not re-test the code as there was no mentioned revision to the software. I went through the README on the github repository and the authors have sufficiently addressed previous comments on the need for more details in the guide.

Reviewer #4 (Remarks to the Author):

I have read the revised version of this manuscript, and I want to applaud the authors for their efforts in revising it. The authors satisfactorily responded to each of my comments and followed my suggestions, and the revised version has improved significantly.

We thank all reviewers again for their constructive feedback and the time they spent reviewing the manuscript.

We numbered all our responses to the reviewers' remarks and addressed them point by point. Changes in the main body of the manuscript are shown in this rebuttal letter and are highlighted in the manuscript itself.

Reviewer #1 (Remarks to the Author):

In the revised manuscript, the authors have provided additional biological interpretations of their results and clarifications on the scientific rationale behind the proposed method. However, the manuscript still lacks convincing evidence that the higher sensitivity of LoDEI is detecting true and biological meaningful editing sites. By aggregating replicates and using a genome-wide false discovery rate calculation approach, the method is potentially detecting a high number of false positives in their newly identified editing sites. If this is the case, LoDEI does not out-perform any of the current published methods and the use of its results is potentially detrimental to a study as it might lead to misinterpretation of biological outcomes. Without any experimental validation, simulation data, or cross validation using additional RNA-seq datasets, the authors have not provided any additional evidence to validate their claim that LoDEI is more accurate than other methods, albeit being able to detect a higher number of editing sites. Overall, the revised manuscript lacks evidence that LoDEI is an improved method for differential editing site detection.

Specific points:

1. The authors should perform experimental validation of their findings using a low throughput method to demonstrate that the method can detect true editing events missed by other methods. Otherwise, the authors should provide some cross-validation of their findings using additional RNA-seq datasets. For example, the authors can show that the detection of new differential editing sites can be reproduced reliably in multiple datasets with the same expected editing outcomes.

1.1 We thank the reviewer for the repeating critique of the used data sets and criticizing the general concept of an empirical estimation of FDRs and q values. It has been validated in the publication of the used ADAR KD data that A-to-I editing is affected directly upon ADAR KD. The general idea of utilizing non-AG mismatches to estimate q -values / FDRs has been used successfully in various studies for detecting A-to-I editing but never in a window-based approach. The general concept of using empirical mismatches from NGS data to approximate FDRs has been used also outside of the field of A-to-I editing. Thus, we are extending a widely accepted procedure. Applying our empirical q value estimation provided by LoDEI to all possible mismatches does not detect any non-A-G mismatches in the ADAR KD and Ro60 KO data sets except a small number of T-C mismatches in the ADAR KD data (Supplementary Table 1) that can be explained by antisense RNAs overlapping with sense transcripts supporting the general idea. The number of differentially edited sites and FDRs aligns with the original ADAR KD publication, further supporting the validity of the results.

2. It has been pointed out by several reviewers that the aggregation of replicates, by taking the mean, leads to the loss of statistical power in detecting differential editing sites as the variance is not considered. This point has not been addressed by the authors. The use of a global q-value is not sufficient when considering highly variable sites, e.g., sequencing artefacts. The authors should address and elaborate on this in detail.

1.2 We thank the reviewer for this comment and are sorry for the misunderstanding. The mean is taken from the sum of all single sites within a window (Fig. 1). Thus, the variability of a single site or, e.g., sequencing artifacts, is treated implicitly by considering sums of sites within a window. As written in the manuscript, the motivation to use a window instead of relying on single sites has also been considered successfully by other A-to-I editing detection approaches for the same reasons.

3. Although the authors show a high degree of overlap between their detected A-to-I editing sites and the REDportal database, they have yet to demonstrate that the additional sensitivity provided by LoDEI has detected new true editing events and not false positives. The high overlap does not address the potential problem of a high false positive rate. In addition, this comparison does not consider differential editing events, which is presented as a main feature of LoDEI, but only the detection of basally edited sites. A visualisation of a select few editing sites in the same dataset that LoDEI was tested on is “double-dipping” and does not suffice as validation of their findings.

1.3 We thank the reviewer for the comment and apologize for not stating the potential limitation that the REDportal database consists of “basally edited sites” rather than differentially edited sites. However, following the reviewer’s arguments would mean that the REDportal database contains many false positives. Instead, we, and the other reviewers, find it supporting if detected differentially edited sites are also listed in a database of “basally edited sites” as provided by the REDportal since differentially edited sites should be - from a theoretical perspective - a subset of “basally edited sites”. A site that is already known to be “basally” edited might be edited differently upon a different condition. We added the potential limitation to the discussion of the main manuscript.

Changes in the discussion section of the revised manuscript (lines 700-703):

“Since the data offered by the REDportal does not concentrate on differentially edited sites, the comparison might be limited in that way. From a theoretical perspective, differentially edited sites should be a subset of generally edited A-to-I sites.”

4. In supp fig. 7 and supp fig. 11, although the authors show that the overlap of the intersection across the tested window sizes are >80%, there appears to be large variability in LoDEI’s sensitivity of detection of differential editing windows. This is potentially a major drawback of the method as its sensitivity can be heavily influenced by the choice of its run parameters. Although the authors have added a comment on how the window size affects the resolution, they have not provided any information or

empirical evidence on their (or the user's) choice of window size. The authors should address this with additional data or a very detailed discussion.

1.4 We thank the reviewer for the comment and apologize for not giving the reader more insights about the effect of chosen parameters. Similar to the discussion of the choice of a q value threshold, the choice of the windows size is a subjective one. The provided analysis that showed an overlap of >80% indicates that the variability between detected windows is neglectable. Thus, the choice of the windows size is to some extent dependent on the user's preference for resolution. Having said that, we think that any biologist who examines a particular window reported by LoDEI will start looking at individual nucleotides under all circumstances independent if the window detected by LoDEI is of size 50 or 100 and will start evaluate sites of interest individually.

Reviewer #2 (Remarks to the Author):

I believe the authors have done a thorough job in addressing my comments (and those of the other reviewers).

One small point relating to my previous comment about sample variability. The authors have clearly demonstrated the utility of their empirical-based method, but I think it is still worth a mention in the Discussion that a limitation is that it does not take into account the sample-sample editing variability within each condition.

2.1 We thank the reviewer for the positive feedback and added the missing point regarding sample-sample variability to the discussion of the manuscript.

Reviewer #3 (Remarks to the Author):

Reviewer #3 (Remarks on code availability):

I did not re-test the code as there was no mentioned revision to the software. I went through the README on the github repository and the authors have sufficiently addressed previous comments on the need for more details in the guide.

3.1 We thank the reviewer for the positive feedback and for reviewing the readme section of the GitHub repository.

Reviewer #4 (Remarks to the Author):

I have read the revised version of this manuscript, and I want to applaud the authors for their efforts in revising it. The authors satisfactorily responded to each of my comments and followed my suggestions, and the revised version has improved significantly.

4.1 We thank the reviewer for the very positive feedback and for investing time to review the revised manuscript.